# Dearomatization drives complexity generation in freshwater organic matter

Siyu Li[1], Mourad Harir[1,2], David Bastviken[3], Philippe Schmitt-Kopplin[1,2], Michael Gonsior[4], Alex Enrich-Prast[3,5], Juliana Valle[1] & Norbert Hertkorn[1,3 ✉]

Dissolved organic matter (DOM) is one of the most complex, dynamic and abundant sources of organic carbon, but its chemical reactivity remains uncertain[1–3]. Greater insights into DOM structural features could facilitate understanding its synthesis, turnover and processing in the global carbon cycle[4,5]. Here we use complementary multiplicity-edited [13]C nuclear magnetic resonance (NMR) spectra to quantify key substructures assembling the carbon skeletons of DOM from four main Amazon rivers and two mid-size Swedish boreal lakes. We find that one type of reaction mechanism, oxidative dearomatization (ODA), widely used in organic synthetic chemistry to create natural product scaffolds[6–10], is probably a key driver for generating structural diversity during processing of DOM that are rich in suitable polyphenolic precursor molecules. Our data suggest a high abundance of tetrahedral quaternary carbons bound to one oxygen and three carbon atoms ($OC_qC_3$ units). These units are rare in common biomolecules but could be readily produced by ODA of lignin-derived and tannin-derived polyphenols. Tautomerization of (poly)phenols by ODA creates non-planar cyclohexadienones, which are subject to immediate and parallel cycloadditions. This combination leads to a proliferation of structural diversity of DOM compounds from early stages of DOM processing, with an increase in oxygenated aliphatic structures. Overall, we propose that ODA is a key reaction mechanism for complexity acceleration in the processing of DOM molecules, creation of new oxygenated aliphatic molecules and that it could be prevalent in nature.

DOM is one of the most complex, dynamic and abundant sources of organic carbon on Earth, and its chemical reactivity remains mysterious so far. The metabolism of autotrophic organisms is well understood and produces a limited number of organic molecules, often rather small biomolecules or polymerized from small repetitive units. Compared with biomolecules, most DOM accumulated in natural waters and soils seem to be extremely complex and rather refractory. The insufficient understanding of the diagenesis of DOM has given rise to many inconclusive hypotheses lacking firm links between biomolecules and the observed DOM molecular complexity.

Aquatic DOM represents a mix of various stages of biotic and abiotic processed terrestrial and aquatic sources across contrasting conditions of temperature, photochemistry and seasonality[2]. Large contrasts of these regimes are observed in tropical and boreal biomes. The Amazon basin is an exemplary tropical catchment and the largest drainage system in the world, responsible for 20% of the global freshwater discharge and for about 10% of the global riverine DOM export to the oceans[11,12]. It comprises heterogeneous landscapes including the Andean Cordillera, minor mountain areas and expansive forested flatlands with stagnant and flowing waters affected by seasonal flooding[13]. The Amazon biome comprises extraordinary biodiversity of plants, animals and microorganisms[14–16], constituting the source of Amazon DOM (AZ-DOM); high temperature combined with high humidity leads to rapid and extensive biological and chemical processing, affecting production and degradation of organic compounds, as well as carbon fluxes[11,17]. The equatorial position of the Amazon ecosystem also promotes photo-oxidation and mineralization of AZ-DOM. Processing of terrigenous and aquatic organic matter in Amazon rivers produces a quarter of global $CO_2$ emissions from inland waters, nearly the same amount of carbon as sequestered by its forest[11,18].

The Amazon basin comprises three main water types. Whitewater rivers (such as the Amazon main course and the Juruá, Japurá, Purus, Solimões and Madeira rivers) are turbid and originate in the Andes, from which they transport large amounts of nutrient-rich sediments[12,19]. Blackwater rivers (such as the Negro River) drain the Precambrian Guiana Shield, carrying small quantities of suspended matter but large amounts of humic substances[20,21]. Clearwater rivers (such as the Tapajós and Xingu rivers) feature high transparency, low sediment load, low nutrients and considerable bacterial abundance[22].

The boreal forest biome is the second largest water-rich landscape apart from the humid tropics, covering about 14% of Earth's land area from 50° N to 70° N, and is associated with forests and wetlands such as bogs, fens and peatlands that store and process vast amounts of carbon. The boreal biome has the largest number of lakes on Earth[23].

[1]Research Unit Analytical Biogeochemistry (BGC), Helmholtz Munich, German Research Center for Environmental Health, Neuherberg, Germany. [2]Chair of Analytical Food Chemistry, Technische Universität München, Freising-Weihenstephan, Germany. [3]Department of Thematic Studies – Environmental Change, Linköping University, Linköping, Sweden. [4]Chesapeake Biological Laboratory, University of Maryland Center for Environmental Science, Solomons, MD, USA. [5]Institute of Marine Science, Federal University of São Paulo, Santos, Brazil. ✉e-mail: Norbert.hertkorn@liu.se

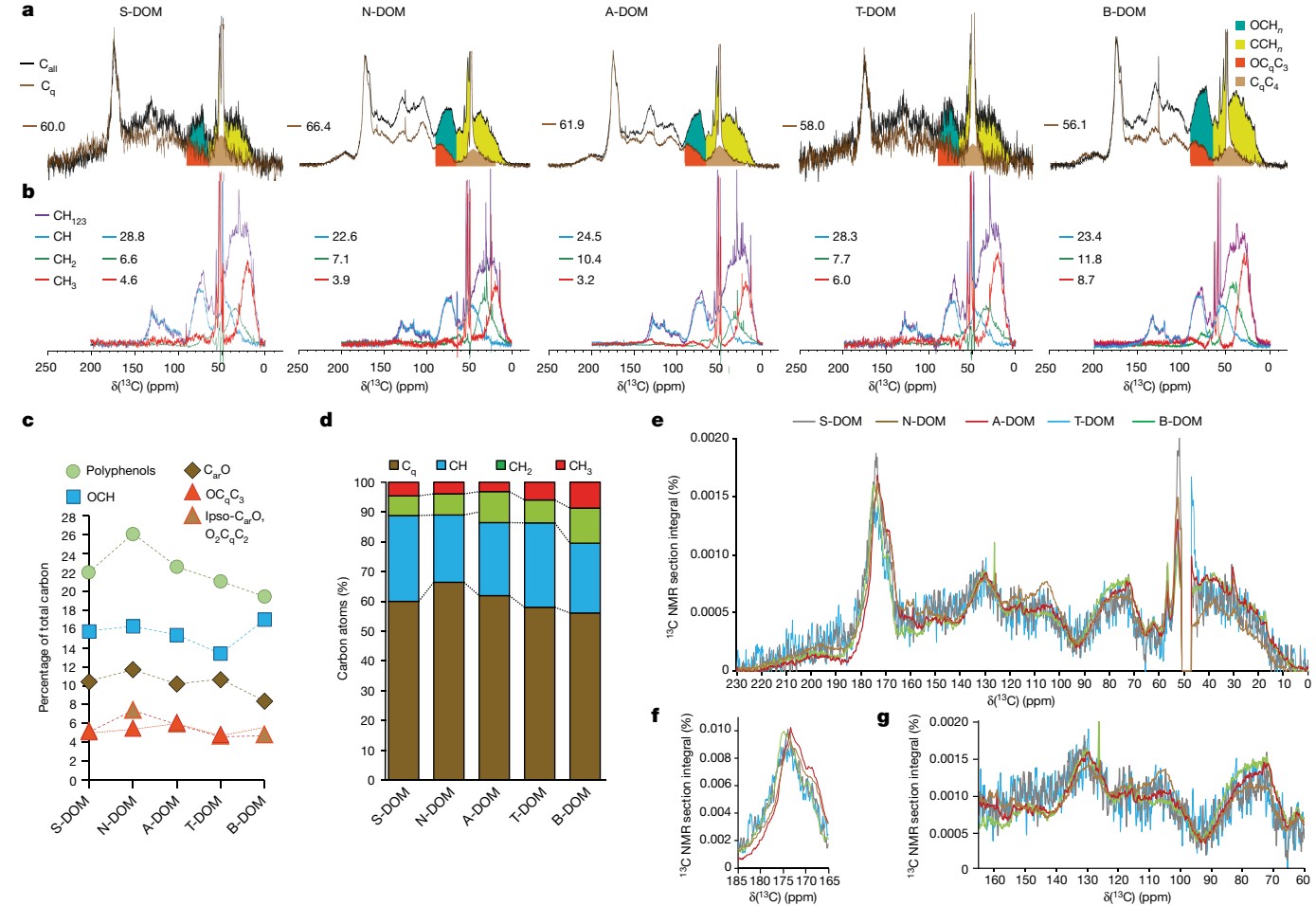

**Fig. 1 | ¹³C NMR spectra of five DOM define contributions of core carbon substructures CH₀₁₂₃. a**, Overlay of single-pulse ($C_{all}$, black) and QUAT ($C_q$, brown) ¹³C NMR spectra; numbers indicate relative proportions of $C_q$ to $C_{all}$ (%); $C_q$-related substructures $OC_qC_3$ and $C_qC_4$, as well as $CH_n$-related substructures $OCH_n$ and $CCH_n$, are shaded in colour. **b**, Overlay of multiplicity-edited ¹³C DEPT NMR spectra, indicating $CH_{123}$ (purple), CH (blue), $CH_2$ (green) and $CH_3$ (red); numbers indicate their relative proportions to $C_{all}$ (%). **c**, Proportions of $OC_qC_3$ and other ODA-relevant oxygenated carbon units to $C_{all}$ (%). **d**, ¹³C NMR-derived

relative proportions of quaternary carbon ($C_q$), methine (CH), methylene ($CH_2$) and methyl ($CH_3$) carbons denote progressive compaction of DOM molecules in the order B-DOM < T-DOM < A-DOM < S-DOM < N-DOM (see text). **e**, Overlay of area-normalized ¹³C NMR spectra of five DOM ($\delta_C$: 0–235 ppm = 100% area). **f**, Overlay of area-normalized ¹³C NMR spectra of five DOM; section of carbonyl derivatives ($\delta_C$: 165–185 ppm = 100% area). **g**, Overlay of area-normalized ¹³C NMR spectra of five DOM; section of polyphenols ($\delta_C$: 60–165 ppm = 100% area).

The molecular composition of boreal lake DOM is considered to be shaped by microbial synthesis and degradation, precipitation, temperature, land cover and water residence time[24–27].

## ¹³C NMR spectra of DOM

Previous mass-spectrometry studies have identified thousands of ions in tropical and boreal DOM and showed distinction of DOM from different waters in the Amazon basin and boreal lakes[17,28–30]. Although high-resolution mass spectrometry offers exceptional capacity to identify elemental compositions and molecular formulae in complex mixtures, such analyses provide very limited specific structural information[31]. NMR spectroscopy offers isotope-specific determination of close-range atomic order (such as for ¹H and ¹³C nuclei) within molecules and standalone capability to explain molecular structures in complex mixtures of unknown molecules such as DOM[32–35]. Here we used complementary multiplicity-edited ¹³C NMR spectra to quantify key substructures assembling the carbon skeletons of DOM in four main Amazon rivers and two mid-size Swedish boreal lakes (Fig. 1, Table 1, Extended Data Figs. 1 and 2 and Extended Data Tables 1 and 2). We have assessed the attendant aspects of DOM formation and reactivity enabled by this in-depth structural analysis.

¹³C NMR spectra detect all carbon atoms in DOM molecules. Combined analysis of multiplicity-edited DEPT (distortionless enhancement by polarization transfer), QUAT (quaternary carbon only) and single-pulse ¹³C NMR spectra ($C_{all}$) provided quantification of all four fundamental chemical environments of quaternary carbon ($C_q$), methine (CH), methylene ($CH_2$) and methyl ($CH_3$) carbon in the five DOM (Fig. 1a,b,d and Extended Data Table 3). These $CH_{0123}$ subspectra showed prominent broad ¹³C NMR resonances representing core-carbon-based structural units of the carbon skeleton of DOM molecules. The ¹³C NMR-derived average O/C (oxygen to carbon) atomic ratios[32] followed the order of N-DOM (DOM in the Negro River) > S-DOM (DOM in the Solimões River) > T-DOM (DOM in the Tapajós River) > B-DOM (DOM in the boreal lakes) > A-DOM (DOM in the Amazonas River). The average H/C (hydrogen to carbon) atomic ratios followed roughly the reverse order N-DOM < T-DOM < S-DOM < A-DOM < B-DOM (Extended Data Table 4), suggesting that the main oxygen-containing functional groups in DOM were associated with unsaturated carbon units, such as $C_{sp2}$-based carbonyls ($C_2C=O$), carbonyl derivatives (CONH, COOH and COOR), oxygenated aromatic carbons ($C_{ar}$–O; polyphenols) and olefins.

B-DOM showed a higher ratio of aliphatic protons to aliphatic carbons compared with the other four DOM, indicating higher H/C ratios within its aliphatic units. The abundance of singly oxygenated aliphatic

## Table 1 | Percentages of 11 [13]C NMR-derived key carbon chemical environments (CH$_{0123}$) in DOM samples

| δ($^{13}$C) (ppm) | Key substructure | S-DOM | N-DOM | A-DOM | T-DOM | B-DOM |
|---|---|---|---|---|---|---|
| 187–235 | C=O | 4.2 | 5.0 | 2.6 | 7.4 | 3.6 |
| 167–187 | COOH | 18.2 | 16.6 | 15.2 | 15.3 | 15.7 |
| 145–167 | C$_{ar}$–O | 10.5 | 11.7 | 10.2 | 10.7 | 8.4 |
| 108–145 | sp$^2$–C$_q$ | 9.5 | 13.1 | 11.0 | 8.4 | 8.7 |
| 108–145 | sp$^2$–CH | 12.8 | 9.9 | 10.4 | 12.6 | 12.5 |
| 90–108 | sp$^2$–C$_{ar}$C (ipso-C)*, O$_2$C$_q$C$_2$ | 5.1 | 7.4 | 5.9 | 4.6 | 4.7 |
| 90–108 | O$_2$CH | 1.7 | 1.3 | 1.2 | 2.7 | 2.0 |
| 47–90 | OC$_q$C$_3$ | 5.0 | 5.4 | 6.0 | 4.7 | 5.6 |
| 66–90 | OCH | 15.8 | 16.3 | 15.4 | 13.4 | 17.0 |
| 20–66 | C$_q$C$_4$ | 9.4 | 5.5 | 6.9 | 9.2 | 5.8 |
| 0–66 | CCH | 7.8 | 7.8 | 15.2 | 10.9 | 16.1 |

C$_{sp3}$-based quaternary carbon C$_q$ units (that is, carbon not carrying any hydrogen according to NMR notation) are: O$_2$C$_q$C$_2$, OC$_q$C$_3$ and C$_q$C$_4$. *Ipso-C$_{ar}$C refers to 1,3,5-trioxo-polyphenols[33].

groups (OCH units) followed the order B-DOM > N-DOM > A-DOM ≈ S-DOM > T-DOM (Table 1); analogous trends applied to the sum of O$_2$CH and OCH units, but highly oxygenated polyphenols in N-DOM and A-DOM contributed to δ$_C$ ≈ 90–108 ppm as well[33]. N-DOM showed the highest proportions and the largest molecular diversity of polyphenolic molecules among the five DOM, covering the maximum [13]C NMR chemical shift range (δ$_C$ ≈ 95–165 ppm) attainable for this class of molecules[33] (Fig. 1c,e,g). Carboxylic acids in DOM (δ$_C$ ≈ 165–185 ppm) showed remarkable variance in abundance (S-DOM > N-DOM > B-DOM ≈ T-DOM ≈ A-DOM) (Table 1 and Extended Data Table 4) and structural diversity (Fig. 1f and Extended Data Fig. 2e), with N-DOM and A-DOM being most distinct. The relative abundance of aliphatic carboxylic acids (δ$_C$ > 175 ppm; Fig. 1f) was lowest in N-DOM and A-DOM. The abundance of ketones in DOM was higher in T-DOM, N-DOM and S-DOM and lower in B-DOM and A-DOM (Table 1, Extended Data Fig. 2 and Extended Data Tables 4 and 5).

## Quaternary carbon is abundant in DOM

The proportion of C$_q$ in total carbons was remarkably high in all five DOM (N-DOM: 66% > A-DOM: 62% > S-DOM: 60% > T-DOM: 58% > B-DOM: 56%), contrasting the comparatively minor fraction of C$_q$ (about 15%) in common, hydrogen-rich primary/central metabolites (Fig. 1d and Extended Data Table 3). C$_q$ in [13]C NMR spectra of all five DOM comprised nine main structural environments (Extended Data Figs. 3 and 4 and Extended Data Table 5). Also, the sum of C$_q$ and CH exceeded 80% of total carbons in all five DOM (Fig. 1d and Extended Data Table 3), indicative of a high degree of compaction and unsaturation of DOM molecules, which is not attainable by any combination of common, hydrogen-rich biomolecules. C$_{sp2}$-based C$_q$ units comprise familiar unsaturated functional groups (that is, C$_2$C = O, COOH, CONH, COOR, C$_{ar}$O and C$_{ar}$C, and C$_2$C = C), resonating at δ$_C$ ≈ 95–235 ppm. The carboxyl group COOH (about 16%) was the most abundant C$_q$-containing functional group in DOM molecules. Moreover, we observed C$_{sp3}$-based C$_q$ units, in particular, OC$_q$C$_3$ (about 6%) and C$_q$C$_4$ (about 7%), resonating at δ$_C$ ≈ 40–110 ppm (Table 1); O$_2$C$_q$C$_2$ units were present but rare.

Mass spectra of tropical riverine and boreal lake DOM showed low average H/C ratios of DOM[17,28–30], and this considerable unsaturation is commonly attributed to the presence of C$_{sp2}$-based hydrogen-deficient structures, such as ketones, carboxylic acids, olefins and polyphenols. Many diverse oxidation processes lead to ketones and carboxylic acids, and riverine DOM typically contain high abundance of polyphenols (Fig. 1, Extended Data Fig. 3 and Extended Data Table 5). However,

compared with the presence of trigonal planar sp$^2$-hybridized C$_q$, the presence of C$_{sp3}$-based tetrahedral C$_q$C$_4$ and OC$_q$C$_3$ units in DOM molecules implies a more stringent and entirely independent structural constraint (Fig. 2, Extended Data Figs. 3 and 4 and Extended Data Table 5). In comparison with the C$_{sp2}$-based structural flatland of single and fused benzene rings[36], C$_q$C$_4$ and OC$_q$C$_3$ units are the ultimate carriers of aliphatic branching and deeply embedded in molecules with complex three-dimensional shapes by necessity. The high abundance of C$_q$C$_4$ and OC$_q$C$_3$ units conveys the characteristics of DOM molecules rich in aliphatic unsaturated structures, such as several fused and bridged alicyclic rings containing several tetrahedral carbon stereocentres. It is worth noting that C$_q$C$_4$ units may originate from many distinct chemical precursors and processes[37], whereas the OC$_q$C$_3$ units have rather limited diversity of sources.

The OC$_q$C$_3$ substructure is very rare in common metabolites; it does not occur in typical carbohydrates, lignins, lipids, nucleotides, peptides and tannins. It was, however, very abundant in all five DOM of this study, comprising up to roughly 6% of all carbon (C$_{all}$), equivalent to about 30% of oxygenated aliphatic (OCH) units (Table 1, Fig. 1c, Extended Data Figs. 3 and 4 and Extended Data Table 5). This mandates mechanistic relevance and straightforward synthesis of OC$_q$C$_3$ units in freshwaters across biomes. In the comparison of all five DOM, OC$_q$C$_3$ units were most abundant in B-DOM and least abundant in T-DOM and N-DOM (Table 1, Fig. 1c, Extended Data Fig. 4 and Extended Data Table 5). Furthermore, we found the abundance of benzene derivatives with electron-donating substituents (-OH and -OCH$_3$; Table 1, Fig. 1c, Extended Data Fig. 4 and Extended Data Table 5) to be highest in N-DOM and lowest in B-DOM.

## ODA creates complexity in freshwater DOM

The high abundance of OC$_q$C$_3$ units in boreal lake and tropical riverine DOM most likely results from ODA of abundant hydroxylated and methoxylated benzene derivatives, which ultimately originate from prevalent and molecularly heterogeneous lignin and tannin degradation products that are common constituents of terrestrial DOM. Phenol, (*para*) 2,5-cyclohexadienone and (*ortho*) 2,4-cyclohexadienone are interconvertible tautomers of C$_6$H$_6$O (Fig. 2a,b), with increasing energy content and reactivity, respectively[38–40]. Resonant electron donation by oxygenated substituents destabilizes benzene rings[39], making them susceptible to transformation into first-generation synthons, comprising masked *ortho*-benzoquinone ketals, *o*-quinols, masked *para*-benzoquinone ketals, *p*-quinols and quinone methides[41–43]. All of those cyclohexadienones are accessible by straightforward reactions from the common aromatic substructures abundant in freshwater DOM (Fig. 2a,b). Cyclohexadienone-based dearomatization is a key biochemical reaction to generate structural complexity and it is also one of the most widely used complexity-generating reactions in organic synthetic chemistry at present to create elaborate natural product scaffolds[6–10,41–48]; here we propose that it is a key environmental mechanism in DOM processing as well. Cyclohexadienones show substituent-dependent atom-specific reactivity at each position of the six-membered rings (that is, substituent-dependent electrophilic and nucleophilic character), setting the stage for a huge variety of follow-up reactions[6,43,49] (Fig. 2c–e). Cyclohexadienones readily engage in, for example, standard and inverse electron-demand Diels–Alder reactions ([4 + 2] cycloadditions), [*m, n*] cycloadditions, cyclizations, additions, reductions and so on, and the initial products often undergo well-documented complementary and parallel cascade reactions[8,50,51]. Already, the basic succession of ODA and [4 + 2] cycloaddition transforms five sp$^2$-hybridized carbon atoms into five sp$^3$-hybridized carbon atoms (Extended Data Table 6).

ODA operates through both biotic and abiotic mechanisms[9,41,50]. Molecular diversification is further amplified through dearomatization by complementary selectivity of its photochemical[52,53], redox-initiated radical[54,55], ionic[56], as well as enzymatic variants; the last of these

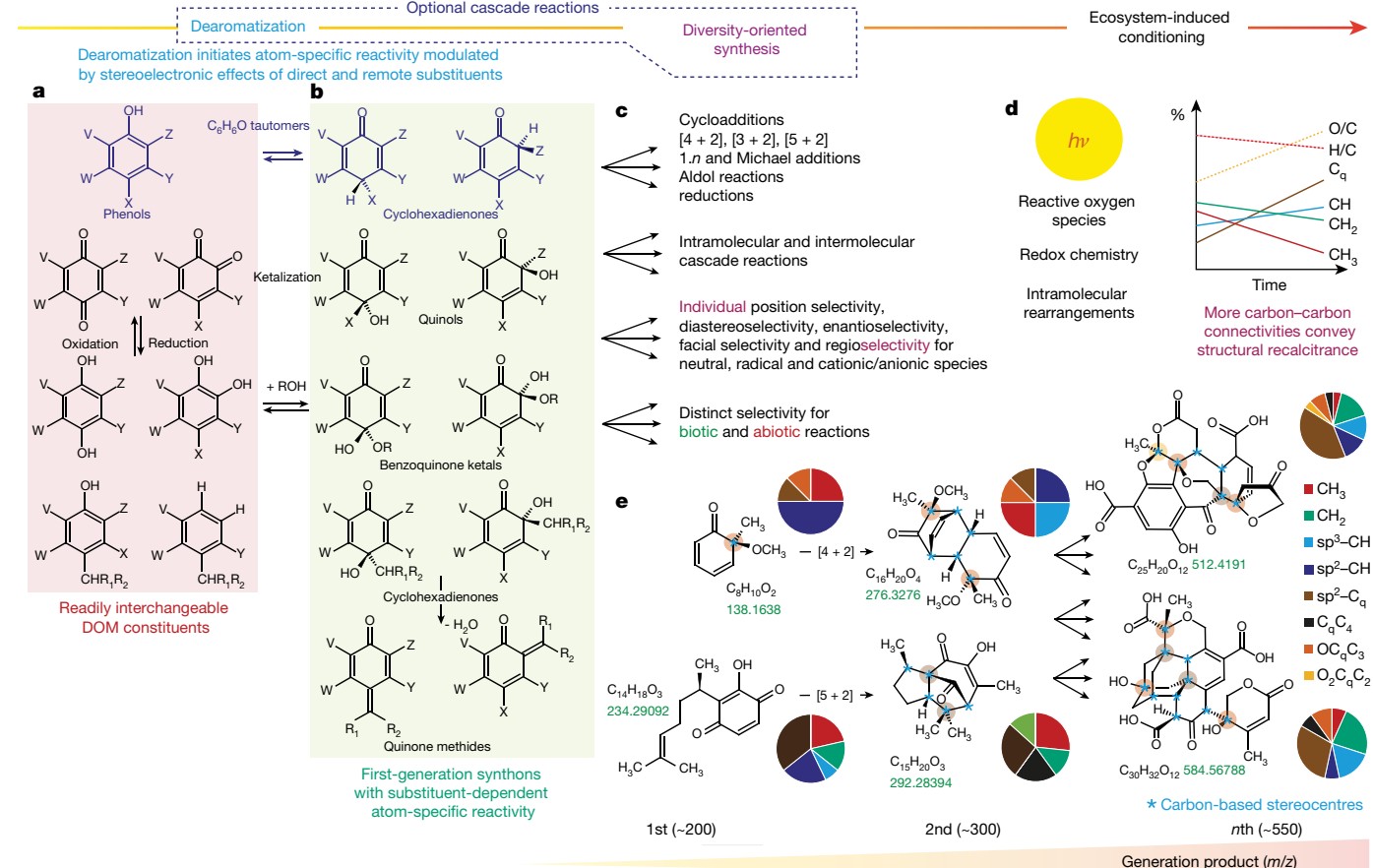

**Fig. 2 | Main synthons and chemical reactions for ODA of DOM.**
**a**,**b**, DOM-related phenolic molecules (red shade) readily convert into five main first-generation (more reactive) *ortho*-cyclohexadienone and (more stable) *para*-cyclohexadienone derivatives (green shade), which possess atom-specific reactivity depending on substitution. **c**, ODA initiates complementary consecutive and parallel reactions that produce a prolific diversity of molecules with elaborate three-dimensional shapes and large-scale obliteration of original binding motifs. **d**, Early-generation products will continually experience

ecosystem-specific transformative conditions and endure exposure to reactive oxygen species, photochemistry and redox chemistry, by which structural recalcitrance of DOM molecules increases with growing counts of intramolecular carbon–carbon bonds. **e**, ODA converts simple cyclohexadienones into complex, oxygen-rich molecules with fused and bridged alicyclic motifs that carry many carbon-based stereocentres, denoted here by blue asterisks; small circles on atoms denote $C_{sp2}$-based units $C_qC_4$, $OC_qC_3$ and $O_2C_qC_2$; large circles denote relative proportions of key carbon units in given molecules.

accommodates a remarkable promiscuity of substrates[10,57,58], fostering opportunities for large-scale DOM processing. All of these dearomatization reactions probably occur in parallel, facilitating rapid complexification within DOM from simple aromatic precursor molecules[8] (Fig. 2 and Extended Data Fig. 5). Fundamentally, ODA transforms flat aromatic rings into elaborately shaped oxygenated aliphatic molecules rich in tetrahedral $C_{sp3}$–carbon atoms with fused and bridged alicyclic rings (Fig. 2e). Aromatic precursor molecules in DOM are often of appreciable size ($m/z \approx 450$ by mass spectrometry)[28,30], polysubstituted, polyoxygenated, molecularly diverse and have inherent low symmetry. ODA chemistry of these molecules will inevitably produce highly complex mixtures of oxygen-rich alicyclic DOM molecules[59–61] (Fig. 2).

We propose ODA chemistry of oxygenated aromatic DOM molecules as an indispensable initiator for the synthesis of $OC_qC_3$-units-containing, highly complex, oxygen-rich alicyclic DOM molecules in tropical and boreal freshwater ecosystems[61]. The molecules generated early by ODA already contain fused and bridged alicyclic rings with several tetrahedral stereogenic centres[59,60], in which many carbons are bonded to several carbons, thereby decreasing the number of chemical bonds between carbon and oxygen atoms on average. This diffuse embedding of oxygen atoms into aliphatic carbon networks is a specific structural feature of freshwater DOM molecules. By contrast, carbon atoms in common metabolites are regularly clustered together, whereas oxygen

atoms are either diluted (as in lipids and peptides) or concentrated (as in carbohydrates).

Two other environmental synthesis pathways to produce $OC_qC_3$ carbon units are known but seem to be of minor relevance compared with ODA. One is selective preservation of $OC_qC_3$ units in precursor (bio) molecules, such as oxygenated terpenoids[62,63]. The other is the unselective attack of energy-rich hydroxyl radicals on DOM molecules[64]. Hydroxylation may also create $OC_qC_3$ carbon units from suitable aliphatic precursors[65]. However, both pathways cause incremental, additive molecular transformations (Extended Data Fig. 5a and Extended Data Table 6) but are not capable of generating topological complexity from structurally simple precursors as realized by ODA[6,8] (Fig. 2e and Extended Data Fig. 5b). The rather diffuse input of $OC_qC_3$ units from highly diverse molecules into the ecosphere caused by these reactions is very likely not competitive with ODA in the molecular transformation of boreal and tropical DOM, in which up to 50% of carbon can be related with structural features susceptible to ODA either as educts (polyphenols) or products ($OC_qC_3$ units) according to [13]C NMR spectra.

## COOH-based rearrangement and ODA synergy

ODA and carboxylic-acid chemistry carry complementary roles in the processing of DOM. Carboxylic groups are the defining feature

of carboxyl-rich alicyclic molecules (CRAM) that are ubiquitous in DOM across water systems[34,66–68]. The near universal presence of large quantities of highly aliphatic CRAM in DOM is difficult to explain by incremental pathways of common microbial or photochemical reactions. ODA fundamentally generates structural complexity of DOM molecules in a few-step cascade reaction (Fig. 2e, Extended Data Fig. 5b and Extended Data Table 6) and we propose carboxylation of ODA products as a straightforward pathway leading to CRAM.

COOH is a highly reactive attachment $C_q$ unit, whereas all other $C_q$ atoms in DOM molecules are connected to two or more carbon atoms. CRAM observance in $^{13}C$ NMR spectra of DOM implies the co-occurrence of (aliphatic and aromatic) carboxylic acids and alicyclic rings in DOM 'on average'. However, the high abundance of both structural units in DOM, and the considerable size of DOM molecules[17,28,30], infers the presence of both substructures in most DOM molecules. The positioning of COOH towards the surface of DOM molecules conveys independent reactivity, including decarboxylative functionalization and carboxylation through complementary neutral, ionic and radical pathways[54–56]. Microbial and abiotic oxidation of DOM uses molecular oxygen and/or reactive oxygen species (Fig. 2d) to generate carboxyl groups[69–71], an efficient processing step of DOM in oxic surface waters.

We propose COOH chemistry as a critical modifier in the structural evolution of DOM towards more compact molecules during environmental processing, which increases the average number of chemical bonds between constituent atoms in DOM molecules and the proportions of quaternary and methine carbon units, at the expense of methylene and methyl units (Fig. 2d). For instance, intermediates produced by decarboxylation carry intrinsic energy fostering structural rearrangements[72]. In particular, free radicals have distinct reactivity, with skeletal rearrangements towards higher compaction supported by the higher stability of sterically crowded radical positions, opposite to common chemistry, in which increasing steric demands (for example, entry of new substituents to pre-existing atomic environments in molecules) are difficult to attain[37,61]. Intramolecular reactions with participation of abundant carboxyl and hydroxyl groups contribute to other compaction of DOM molecules by, for example, forming anhydrides (two COOH groups), lactones (COOH and OH units) and ethers (two OH units).

Common aquatic DOM contains fewer N-containing or S-containing functional groups than O-containing functional groups[28–32], and their effects on overall $^{13}C$ NMR properties remain limited. However, the dearomatization of precursors such as pyrrole, pyridine, indole and aniline derivatives readily generate alkaloid-like structurally elaborate CHNO molecules under boreal and tropical catchment conditions[73], which could be a main constituent of freshwater CHNO compounds in DOM. Such reactions agree with a recently described prevalence of heterocyclic nitrogen in aged ocean dissolved organic nitrogen[74].

Fundamental structural rearrangement, many carbon–carbon connectivities in hydrogen-deficient molecules and large-scale obliteration of standard biomolecular structural motifs favour intrinsic structural recalcitrance of DOM against expedient degradation. Therefore, small units such as $CO_2$ and $CH_4$ are more likely to be lost than large substructures during the process of DOM molecular evolution. ODA readily explains the observed ultimate structural diversity of DOM molecules and the difficulty in regenerating sizeable amounts of standard biomolecular binding motifs such as simple carbohydrates or amino acids already from early stages of DOM diagenesis because they tend to be lost early[75].

DOM molecules generated from low-mass and high-mass and low-symmetry oxygenated aromatic educts through ODA show elaborate shapes with a large proportion of $sp^3$-hybridized carbon, fused and bridged alicyclic rings, presence of chiral carbon atoms and oxygen-based and nitrogen-based functionalization—features that correlate with success in medical drug design[8,9,36] (Fig. 2e). Architecturally multiform molecules explore larger regions of the chemical space and, when featuring low counts of freely rotating bonds, convey more specific ligand–receptor interactions than flat (aromatic) molecules[76,77]. DOM, a globally relevant layer of ultimate organic molecular complexity, comprises hundreds of gigatonnes of organic carbon, several orders of magnitude more abundant than known biologically active natural products. It is conceivable that some of these polyfunctional, elaborately shaped, compact molecules carry relevant but as yet unrecognized biological activity.

## Conclusions

Polyphenol chemistry in DOM processing comprises a remarkable dichotomy of traditional ring-opening and substitution chemistry on one hand and dearomatization on the other hand. ODA initiates an inflationary increase of molecular structural diversity from early stages of DOM processing, fundamentally distinct from the rather incremental variance in molecular structures associated with the addition and release of small units such as, for example, $\pm H_2$, $CH_2$, O, CO and $CO_2$.

The NMR-based structural differences of boreal lake and tropical river DOM molecules were not larger than the distinction among the four investigated AZ-DOM despite experiencing contrasting regimes of microbial communities, photochemistry, temperature and seasonality during their synthesis and degradation. The proposed ODA pathway applies to both biomes and offers a new mechanism to better reveal, understand and predict DOM structural complexity. It seems that ODA is an important mechanism to produce structurally altered DOM molecules that resist degradation and persist in the environment for centuries to millennia. We suggest that ODA might be a key process in the formation of CRAM that are abundant in freshwater and the global ocean[32,34,68]. It has been shown that CRAM in the deep ocean is old and very resistant to microbial and photochemical degradation[78,79], and sequestration of carbon in structurally recalcitrant CRAM would reduce the release of $CO_2$ to the atmosphere, thereby affecting global warming and climate change. This research opens doors towards more comprehensive understanding of the roles of DOM in ecosystems and as a potential chemical resource to society.

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

## Methods

### Sampling and site locations

39 Amazon basin water samples from 34 sampling sites were collected between 2 April 2014 and 25 May 2014 eastward from Solimões River (whitewater), Negro River (blackwater), Amazonas River (turbid water) to the Tapajós River (clearwater). Water samples were collected during a high water period with unusually high levels of flooding. Ar1–Ar4 were sampled six weeks later than the other Amazonas River samples (Extended Data Fig. 1 (map) and Extended Data Table 1). We obtained water samples by boat just below the surface. Solid-phase extraction (SPE) of the water samples was performed within 2 h in the field. The water column DOM was extracted by a previously described SPE method using PPL resin[29,80,81]. The eluates were stored in the freezer (−20 °C) until further analysis. To obtain meaningful S/N ratios in NMR spectra, we have used four consolidated Amazon basin rivers samples (SNAT) according to water types and selected samples with a very high similarity of their [1]H NMR spectra (data not shown). About 75% of individual samples were used for pooling, after full NMR and mass spectrometry and chemistry characterization (data not used here), leaving backup samples in case of need. The pooling conforms to the aim of this contribution, which attempts the depiction of average structural features of DOM molecules in the four main selected Amazon basin rivers. Swedish boreal lake water samples were collected in August 2012 in the Malingsbo region and two representative lakes were included in this study, namely, Lilla Sångaren (M5) and Övre Skärsjön (M10); isolation of SPE-DOM in Swedish lakes was performed analogous to Amazon river basin waters. M5 and M10 are mid-size boreal Swedish lakes with the following key parameters: dissolved organic carbon: 6.8 and 11.2 mg $l^{-1}$; lake area: 24 and 165 ha; maximum depth: 20 and 32 m; computed water residence time: 1.18 and 1.63 years (ref. 82); averaged values for very similar [13]C NMR spectra of boreal lakes M5 to M10 produced values of B-DOM as shown.

### NMR spectroscopy

A Bruker Avance III spectrometer and TopSpin 3.6/PL6 software were used to acquire [13]C NMR spectra of re-dissolved AZ-DOM (10–40 mg solid SPE-DOM in typically 75–135 μl $CD_3OD$ (99.95% [2]H; [13]C-depleted [12]$CD_3OD$; Aldrich, Steinheim, Germany) at 283 K. Briefly, the re-dissolved DOM were transferred to 2.5–3.0-mm Bruker Match tubes and sealed. A cryogenic classical geometry 5 mm $z$-gradient [13]C, [1]H probe ($B_0 = 11.7$ T) was used for acquisition of [13]C NMR spectra. Transmitter pulses were at approximately 10 μs for [1]H and [13]C and calibrated 90°/180° pulses were used for each sample. In independent experiments, one-dimensional 800 MHz [1]H NMR spectra were acquired from all 39 AZ-DOM samples (data not shown) and the samples showing the most congruent curvature of their [1]H NMR spectra across the entire region of chemical shift ($\delta_H \approx 0$–10 ppm) were pooled before the acquisition of spectra for the four AZ-DOM samples (that is, S-DOM, N-DOM, A-DOM and T-DOM; see Extended Data Table 1); about 75% of samples were used for pooling (Extended Data Table 1) and the residue was kept for eventual consecutive analysis (data not shown). Pooling was necessary to obtain high-quality [13]C NMR spectra ([13]C receptivity $\approx 1.7 \times 10^{-4}$ of [1]H) with sufficient S/N ratio to faithfully resolve low-abundance $C_{sp2}$-based chemical environments. Swedish lake water samples M5 and M10 were used as isolated for acquisition of NMR spectra because of the higher disposable amount of sample; [13]C NMR spectra shown represent M5 (Fig. 1 and Extended Data Fig. 3), but all NMR section integrals and intensity computations of B-DOM represent averaged values of M5 and M10; [13]C NMR spectra of M5 and M10 in essence coincided, but that of M5 showed considerably better S/N ratio than that of M10. We used inverse-gated [1]H decoupling for [13]C NMR spectra to eliminate nuclear Overhauser effects and (acquisition-time-adjusted) linear combinations of the [13]C DEPT-45, DEPT-135 and DEPT-90 NMR spectra ([1]$J_{CH}$: 150 Hz) to compute the individual traces of CH ([13]C DEPT-90 NMR spectrum), $CH_2$ ([13]C DEPT-45 minus [13]C DEPT-135) and $CH_3$ (([13]C DEPT-45 plus [13]C DEPT-135) minus [13]C DEPT-90). We corrected the [13]C DEPT-90 NMR spectrum by subtracting

an appropriate amount (commonly about 2–3%) of the [13]C DEPT-45 NMR spectrum to attenuate leakage of $CH_3$ and $CH_2$ into the [13]C DEPT-90 NMR spectrum (methine carbon (CH) in DOM does not show appreciable [13]C NMR resonances at $\delta_C < 20$ ppm) that arises from the unavoidable variance in [1]$J_{CH}$ of DOM. Then we determined the relative contributions of the individual spectra ($CH_3$, $CH_2$, $CH_1$) to the sum $CH_{123}$ as observed in [13]C DEPT-45 NMR spectra with recognition of the individual transfer amplitudes, which were as follows ($CH_3 = 1.06$; $CH_2 = 1.0$; $CH = 0.707$)[83,84]. The proportions of quaternary carbon atoms $C_q$ in DOM were computed from comparison of [13]C DEPT-45, [13]C QUAT and single-pulse [13]C NMR spectra.

[13]C NMR section integrals and overlay figures were computed using the Bruker AMIX software (version 3.9.4) from area-normalized spectra with 0.1-ppm buckets and 100% total NMR integral area from $\delta_C = 0$–235 ppm, with exclusion of [13]$CD_3OD$, $\delta_{13C} = 47$–51 ppm. We used bucketed [13]C NMR section integral values with 1-ppm bandwidth from $\delta_C = 0$–235 ppm for $C_{all}$ and $C_q$, a bandwidth from $\delta_C = 0$–200 ppm for CH, a bandwidth from $\delta_C = 0$–100 ppm for $CH_2$ and a bandwidth from $\delta_C = 0$–70 ppm for $CH_3$ carbon units, and we set all negative values to zero. By these means, we avoided that baseline drift would influence $CH_{123}$ values at values of $\delta_C$ for which no actual [13]C NMR resonance integral was expected.

The content of polyphenols in [13]C NMR spectra[33] (Fig. 1c) was computed as the sum of $C_{ar}O$ (80%), $C_{ar,q}$ (60%), $C_{ar}H$ (30%) and ipso-$C_{ar,q}$ (80% of integral; see Table 1). See Extended Data Table 2 for further acquisition parameters.

H/C and O/C elemental ratios were computed according to Hertkorn et al.[32] and Fig. 19 in ref. 27.

## Data availability

All data are available in the manuscript, in Dryad at https://doi.org/10.5061/dryad.jsxksn0hr ([13]C NMR data for the five dissolved organic matter) or the Extended Data.

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

**Acknowledgements** This work was supported by the Alexander von Humboldt Foundation (N.H. and A.E.-P.; Research Linkage Program: Connecting the diversity of DOM and $CO_2$ and $CH_4$ production in tropical lakes); the Swedish Research Council for Sustainable Development, FORMAS, grant no. 2021-0242, the Swedish Research Council (VR) (D.B.; grant no. 2012-00048), the European Research Council ERC (D.B.; grant no. 725546), the Chinese Scholarship Council (S.L.; grant no. 201806360268) Molecular Level Characterization of Natural Organic Matter in Tropical Freshwater and Marine Ecosystems of Worldwide Relevance: Amazon Basin, Pantanal, Red Sea, and several grants from Brazilian Foundations CAPES (Probal), FAPERJ (Cientista do Nosso Estado) and CNPq (Universal and Ciencias sem Fronteiras) to A.E.-P. J.V. thanks Brazilian Agency CNPq for a fellowship (290004/2014-4).

**Author contributions** D.B., A.E.-P., M.G., P.S.-K. and N.H. designed the research. D.B., A.E.-P. and N.H. obtained funding. J.V., M.H., D.B., P.S.-K., M.G., A.E.-P. and N.H. participated in sampling in several expeditions from 2012 to 2017. S.L., J.V. and N.H. prepared samples for NMR. S.L. and N.H. acquired NMR spectra. S.L., M.H. and N.H. computed NMR-based relationships. S.L. and N.H. interpreted NMR spectra. All authors contributed in-depth assessment of riverine organic matter processing mechanisms, discussion and writing of the manuscript.

**Funding** Open access funding provided by Linköping University.

**Competing interests** The authors declare no competing interests.

**Additional information**
**Correspondence and requests for materials** should be addressed to Norbert Hertkorn.

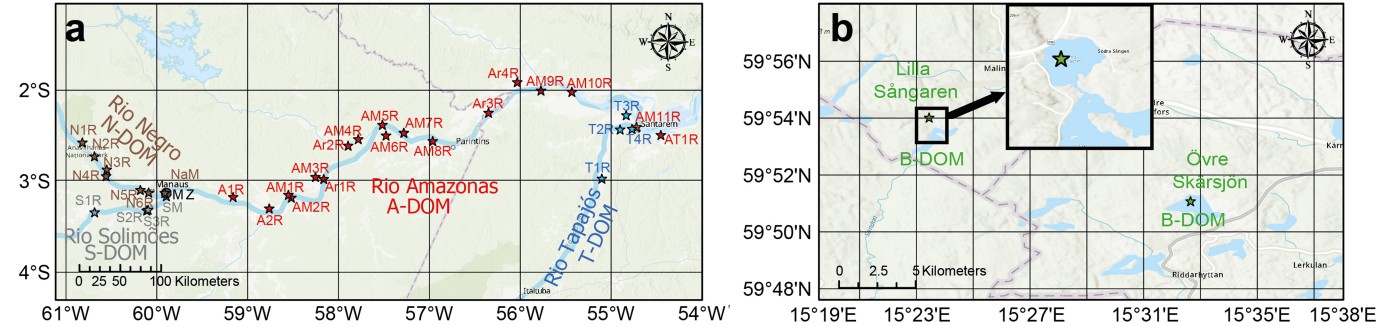

**Extended Data Fig. 1 | Map of sampling locations. a**, Amazon rivers. **b**, Swedish boreal lakes. For annotation and consolidation, see Extended Data Table 1. Amazon basin samples comprised pooled grey (S-DOM), brown (N-DOM), red (A-DOM) and blue (T-DOM) samples, with about 75% of individual samples used for consolidation; asterisk in Extended Data Table 1 denotes projected mass of sample.

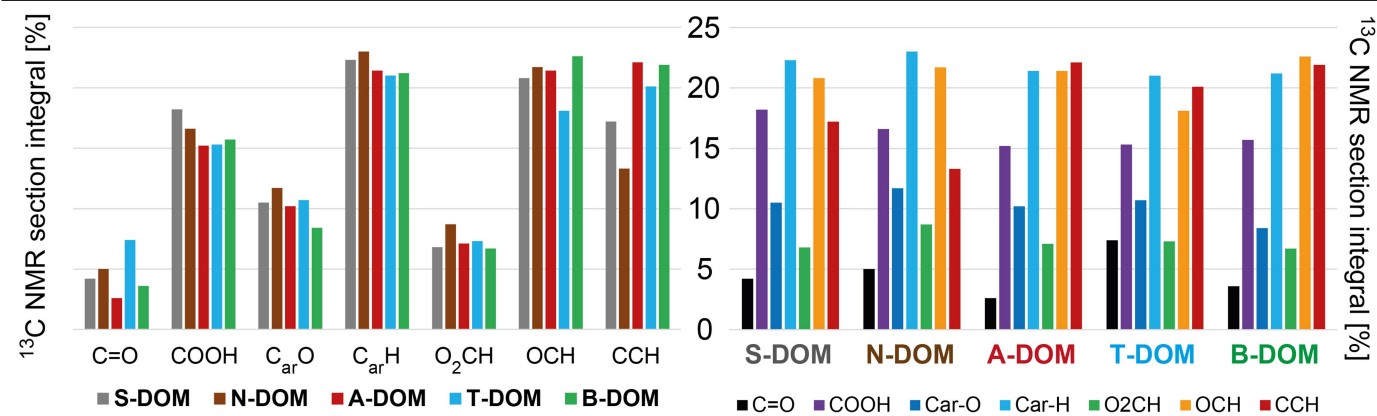

**Extended Data Fig. 2 | Relative proportions of seven fundamental carbon-based substructures in $^{13}$C NMR spectra of five DOM.** Left, sorted according to DOM; right, sorted according to carbon units.

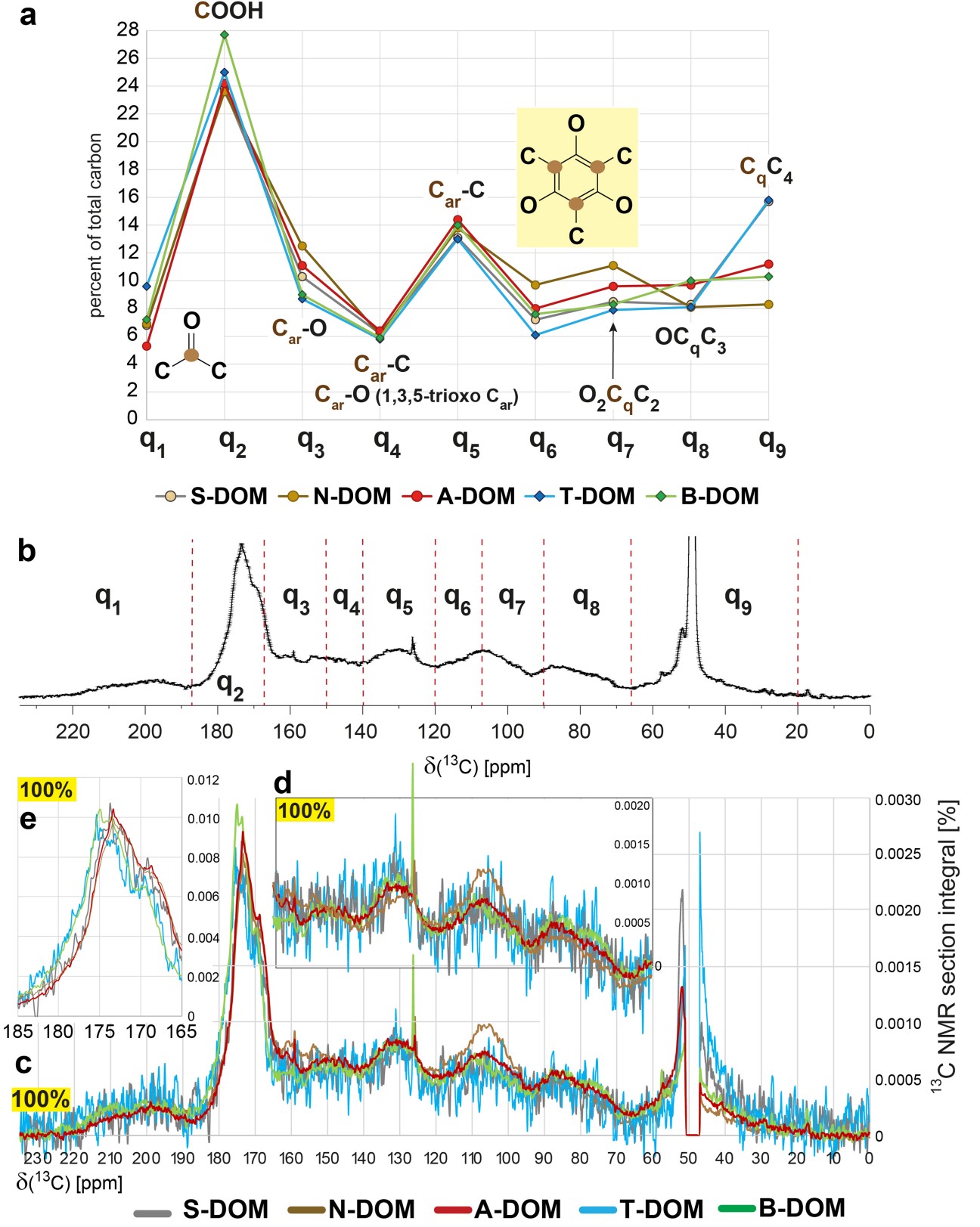

**Extended Data Fig. 3 | Quaternary carbon $C_q$ in five DOM. a**, Relative abundance of nine groups of $C_{sp2}$-based and $C_{sp3}$-based quaternary carbon $C_q$ (that is, carbon not carrying any hydrogen according to NMR notation) in five DOM (SNATB) as indicated (see Extended Data Table 4). **b**, Consolidated QUAT $^{13}$C NMR spectra of five DOM (SNATB) provide excellent S/N ratio to facilitate recognition of $C_q$ units. **c**, Overlay of area-normalized QUAT $^{13}$C NMR spectra of individual DOM. **d,e**, Overlay of area-normalized QUAT $^{13}$C NMR spectra of individual DOM within sections shown (Fig. 1e–g enlarged).

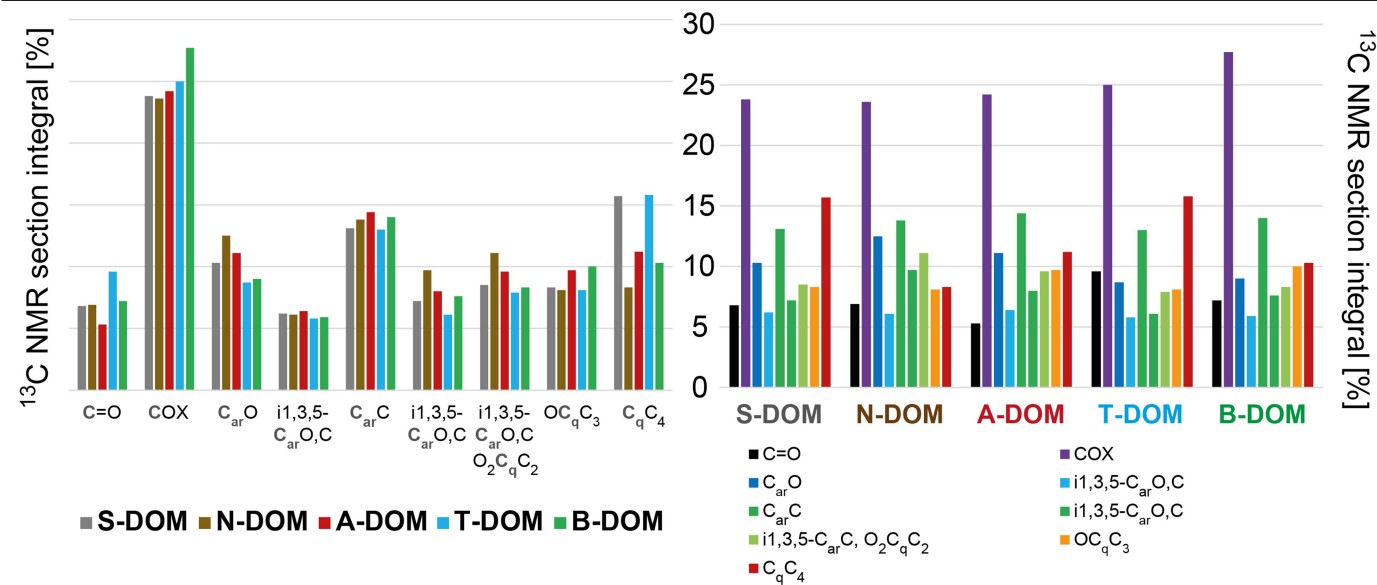

**Extended Data Fig. 4 | Quaternary carbon $C_q$ in DOM.** $^{13}$C NMR-derived proportions of $C_{sp2}$-based and $C_{sp3}$-based quaternary carbon $C_q$ (that is, carbon not carrying any hydrogen according to NMR notation) in five DOM (SNATB), normalized to 100% total $^{13}$C NMR integral of $C_q$ units.

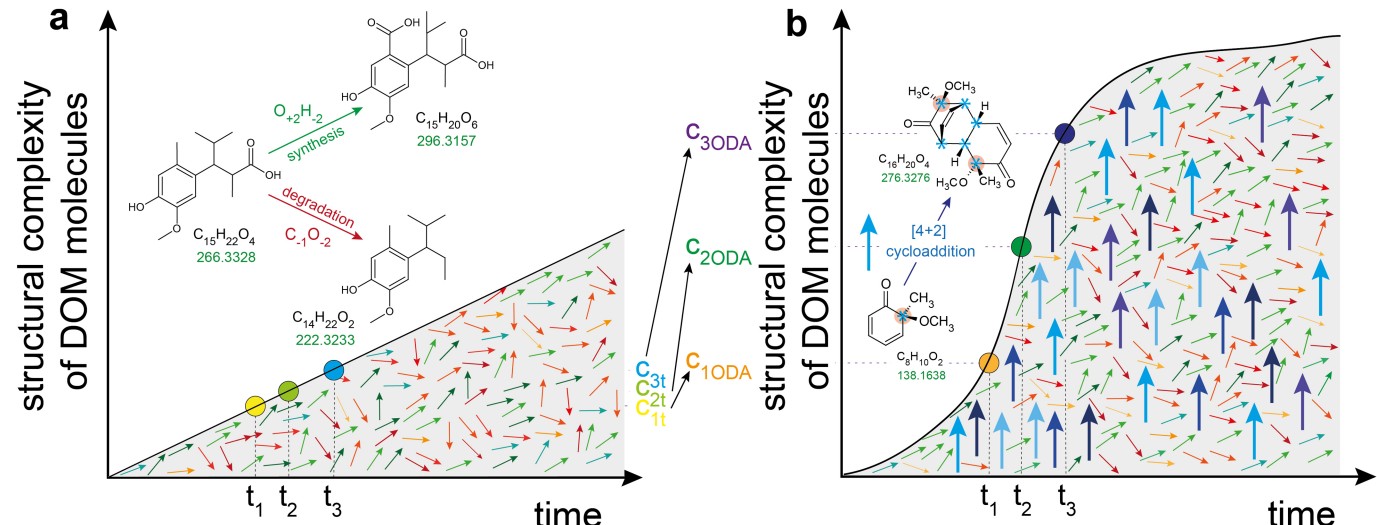

**Extended Data Fig. 5 | Traditional and new view of complexity generation in the processing of DOM molecules.** Both biotic and abiotic transformations affect the chemical structures of DOM molecules, and the consolidated effects of all these reactions define their temporal evolution in the environment. **a**, The present view attributes the temporal evolution of DOM molecules to many successive incremental chemical transformations (items 1–12 in Extended Data Table 6). These commonly refer to gain or loss of small units such as, for example, $\pm H_2$, $CH_2$, O, CO and $CO_2$, and often affect functional groups rather than the carbon skeleton of DOM molecules. Compositional alterations allow less far-reaching conclusions about changes in molecular structures than $^{13}C$ NMR spectra, which detect carbon atomic environments. Synthesis (green arrows) commonly increases molecular complexity, whereas degradation (red arrows) may increase or decrease molecular diversity of DOM. Synthesis here refers to ways by which organic molecules are synthesized in anabolic metabolism, whereas degradation represents catabolic or abiotic degradation of organic matter. Biological processes behind these processes include primary production, secondary production, grazing, excretion, exudation, viral lysis and biodegradation, shaped by community structure and affected by promiscuous enzymes. Abiotic processes, including photochemical and thermal transformations, sorption/desorption and redox reactions, also contribute to these processes. **b**, The new view implies that polyphenol-dependent ODA and consecutive cascade reactions cause fast, inflationary increase of molecular diversity with thorough changes of molecular structures towards complex aliphatic oxygenated molecules from early stages of DOM processing (items 13–16 in Extended Data Table 6). In nature, the classical incremental molecular evolution of DOM operates on top of the ODA chemistry.

**Extended Data Table 1 | Description of investigated sites and sampling information**

| River | Sample name | Longitude (°W) | Latitude (°S) | Sampling date | Water type | Solid weight of NMR sample (mg) |
|---|---|---|---|---|---|---|
| S | S1R | -60.6845 | -3.3414 | 06.04.2014 | white | 1.3 |
| S | S2R | -60.1116 | -3.3209 | 10.04.2014 | white | 3.0 |
| S | S3R | -60.0957 | -3.3136 | 10.04.2014 | white | 3.5 |
| S | SM | -59.8949 | -3.1647 | 29.04.2014 | white | 2.7 |
| **S-DOM** | consolidated sample | | | | | 10.7* |
| N | N1R | -60.8198 | -2.5740 | 20.04.2014 | black | 0.6 |
| N | N2R1 | -60.6865 | -2.7267 | 17.04.2014 | black | 2.6 |
| N | N2R2 | -60.6865 | -2.7267 | 17.04.2014 | black | 3.7 |
| N | N3R | -60.5509 | -2.8719 | 25.04.2014 | black | 2.1 |
| N | N4R1 | -60.5583 | -2.9440 | 21.04.2014 | black | 1.9 |
| N | N4R2 | -60.5583 | -2.9440 | 21.04.2014 | black | 2.0 |
| N | N5R | -60.1810 | -3.0958 | 28.04.2014 | black | 2.4 |
| N | N6R1 | -60.0902 | -3.1242 | 14.04.2014 | black | 1.7 |
| N | N6R2 | -60.0902 | -3.1242 | 14.04.2014 | black | 1.5 |
| N | N6R3 | -60.0902 | -3.1242 | 14.04.2014 | black | 2.6 |
| N | NaM | -59.9126 | -3.1292 | 29.04.2014 | black | 3.1 |
| SN mixing | MZ | -59.8958 | -3.1301 | 29.04.2014 | turbid | 2.8 |
| **N-DOM** | consolidated sample | | | | | 20.2* |
| A | A1R | -60.1596 | -2.7933 | 29.04.2014 | turbid | 3.9 |
| A | A2R | -58.7621 | -3.2992 | 29.04.2014 | turbid | 2.2 |
| A | AM1R | -58.5192 | -3.1802 | 29.04.2014 | turbid | 4.8 |
| A | AM2R | -58.5513 | -3.1492 | 30.04.2014 | turbid | 3.7 |
| A | AM3R | -58.2533 | -2.9530 | 30.04.2014 | turbid | 3.1 |
| A | AM4R | -57.8941 | -2.6091 | 30.04.2014 | turbid | 3.6 |
| A | AM5R | -57.5167 | -2.3797 | 01.05.2014 | turbid | 2.6 |
| A | AM6R | -57.4771 | -2.4972 | 03.05.2014 | turbid | 3.1 |
| A | AM7R | -57.2811 | -2.4676 | 03.05.2014 | turbid | 3.4 |
| A | AM8R | -57.2811 | -2.4676 | 03.05.2014 | turbid | 3.1 |
| A | AM9R | -55.7717 | -2.0033 | 03.05.2014 | turbid | 3.0 |
| A | AM10R | -55.4316 | -2.0180 | 03.05.2014 | turbid | 3.4 |
| A | AM11R | -54.7185 | -2.4041 | 18.05.2014 | turbid | 2.2 |
| A | Ar1R | -58.1626 | -2.9750 | 22.05.2014 | turbid | 2.5 |
| A | Ar2R | -57.7824 | -2.5369 | 22.05.2014 | turbid | 2.5 |
| A | Ar3R | -56.3486 | -2.2473 | 21.05.2014 | turbid | 2.3 |
| A | Ar4R | -56.0326 | -1.9115 | 21.05.2014 | turbid | 3.3 |
| A | AT1R | -54.4518 | -2.4916 | 07.05.2014 | turbid | 1.1 |
| **A-DOM** | consolidated sample | | | | | 40.4* |
| T | T1R1 | -55.1013 | -2.9714 | 15.05.2014 | clear | 2.8 |
| T | T1R2 | -55.1013 | -2.9714 | 15.05.2014 | clear | 2.7 |
| T | T2R | -54.9001 | -2.4315 | 17.05.2014 | clear | 2.1 |
| T | T3R | -54.8332 | -2.2742 | 18.05.2014 | clear | 3.4 |
| T | T4R | -54.7219 | -2.4056 | 18.05.2014 | clear | 3.3 |
| **T-DOM** | consolidated sample | | | | | 10.7* |
| B | M05 | 15.3908 | 59.9003 | 30.08.2012 | boreal lake | 22 |
| B | M10 | 15.5442 | 59.8513 | 31.08.2012 | boreal lake | 25 |
| **B-DOM** | mathematically consolidated sample | | | | | 22 |

**Extended Data Table 2 | Acquisition parameters of NMR spectra**

| Spectrum | Sample | Figure | NS | AQ [ms] | D1 [ms] | WDW2 | PR2 |
|---|---|---|---|---|---|---|---|
| $^{13}$C NMR | S-DOM | 1aefg | 59392 | 1000 | 19000 | EM | 12.5 |
| $^{13}$C NMR | N-DOM | 1aefg | 54791 | 1000 | 19000 | EM | 12.5 |
| $^{13}$C NMR | A-DOM | 1aefg | 60416 | 1000 | 19000 | EM | 12.5 |
| $^{13}$C NMR | T-DOM | 1aefg | 110612* | 1000 | 19000 | EM | 12.5 |
| $^{13}$C NMR | B-DOM | 1aefg | 15104 | 1000 | 19000 | EM | 12.5 |
| $^{13}$C DEPT-45 | S-DOM | 1b | 98304* | 1000 | 2000 | EM | 12.5 |
| $^{13}$C DEPT-45 | N-DOM | 1b | 73728 | 1000 | 2000 | EM | 12.5 |
| $^{13}$C DEPT-45 | A-DOM | 1b | 73728 | 1000 | 2000 | EM | 12.5 |
| $^{13}$C DEPT-45 | T-DOM | 1b | 157357* | 1000 | 2000 | EM | 12.5 |
| $^{13}$C DEPT-45 | B-DOM | 1b | 25600 | 1000 | 2000 | EM | 12.5 |
| $^{13}$C DEPT-135 | S-DOM | 1b | 98304 | 1000 | 2000 | EM | 12.5 |
| $^{13}$C DEPT-135 | N-DOM | 1b | 73728 | 1000 | 2000 | EM | 12.5 |
| $^{13}$C DEPT-135 | A-DOM | 1b | 73728* | 1000 | 2000 | EM | 12.5 |
| $^{13}$C DEPT-135 | T-DOM | 1b | 98304* | 1000 | 2000 | EM | 12.5 |
| $^{13}$C DEPT-135 | B-DOM | 1b | 25600 | 1000 | 2000 | EM | 12.5 |
| $^{13}$C DEPT-90 | S-DOM | 1b | 80765 | 1000 | 2000 | EM | 12.5 |
| $^{13}$C DEPT-90 | N-DOM | 1b | 81469 | 1000 | 2000 | EM | 12.5 |
| $^{13}$C DEPT-90 | A-DOM | 1b | 123904 | 1000 | 2000 | EM | 12.5 |
| $^{13}$C DEPT-90 | T-DOM | 1b | 194560* | 1000 | 2000 | EM | 12.5 |
| $^{13}$C DEPT-90 | B-DOM | 1b | 29120 | 1000 | 2000 | EM | 12.5 |
| $^{13}$C QUAT NMR | S-DOM | 1a | 55296 | 1000 | 19000 | EM | 12.5 |
| $^{13}$C QUAT NMR | N-DOM | 1a | 59336 | 1000 | 19000 | EM | 12.5 |
| $^{13}$C QUAT NMR | A-DOM | 1a | 59005 | 1000 | 19000 | EM | 12.5 |
| $^{13}$C QUAT NMR | T-DOM | 1a | 102181* | 1000 | 19000 | EM | 12.5 |
| $^{13}$C QUAT NMR | B-DOM | 1a | 83203* | 1000 | 19000 | EM | 12.5 |

A cryogenic classical geometry 5 mm $z$-gradient $^{13}$C/$^{1}$H probehead was used for acquisition of NMR spectra ($B_0$ = 11.7 T); NS, number of scans; AQ, acquisition time (ms); D1, relaxation delay (ms); WDW2, apodization functions in F2; EM, line-broadening factor (Hz); PR2, coefficients used for windowing functions WDW2. *Co-added $^{13}$C NMR spectra.

**Extended Data Table 3 | Percentage of multiplicity-edited carbon atomic units $CH_{0123}$ in five DOM (see Fig. 1c)**

| DOM | $C_q$ | $CH_1$ | $CH_2$ | $CH_3$ |
|---|---|---|---|---|
| S-DOM | 60.0 | 28.8 | 6.6 | 4.6 |
| N-DOM | 66.4 | 22.6 | 7.1 | 3.9 |
| A-DOM | 61.9 | 24.5 | 10.4 | 3.2 |
| T-DOM | 58.0 | 28.3 | 7.7 | 6.0 |
| B-DOM | 56.1 | 23.4 | 11.8 | 8.7 |

**Extended Data Table 4 | Relative proportions of seven fundamental carbon-based substructures in $^{13}$C NMR spectra of five DOM**

| δ($^{13}$C) [ppm] | key substructure | S-DOM | N-DOM | A-DOM | T-DOM | B-DOM | H/C ratio | O/C ratio |
|---|---|---|---|---|---|---|---|---|
| 187-235 ppm | C=O | 4.2 | 5.0 | 2.6 | 7.4 | 3.6 | 0 | 1 |
| 167-187 ppm | COOH | 18.2 | 16.6 | 15.2 | 15.3 | 15.7 | 1 | 2 |
| 145-167 ppm | $C_{ar}$-O | 10.5 | 11.7 | 10.2 | 10.7 | 8.4 | 0 | 1 |
| 108-145 ppm | $C_{ar}$-H* | 22.3 | 23.0 | 21.4 | 21.0 | 21.2 | 1 | 0 |
| 90-108 ppm | $O_2CH$* | 6.8 | 8.7 | 7.1 | 7.3 | 6.7 | 1 | 2 |
| 47-90 ppm | OCH* | 20.8 | 21.7 | 21.4 | 18.1 | 22.6 | 1 | 1 |
| 0-47 ppm | CCH* | 17.2 | 13.3 | 22.1 | 20.1 | 21.9 | 2 | 0 |
|  | H/C ratio | 1.025 | 0.966 | 1.093 | 1.019 | 1.100 | | |
| | O/C ratio | 0.855 | 0.890 | 0.788 | 0.814 | 0.794 | | |

*Contains some $C_q$ units; see Extended Data Fig. 2 and Extended Data Table 4.

**Extended Data Table 5 | Quaternary carbon $C_q$ in five DOM**

| notation Fig. S3b | $\delta(^{13}C)$ [ppm] | key substructure | S-DOM | N-DOM | A-DOM | T-DOM | B-DOM |
|---|---|---|---|---|---|---|---|
| $q_1$ | 187-235 ppm | $\underline{C}$=O | 6.8 | 6.9 | 5.3 | 9.6 | 7.2 |
| $q_2$ | 167-187 ppm | $\underline{C}$OOH, $\underline{C}$OOR, $\underline{C}$ONH | 23.8 | 23.6 | 24.2 | 25.0 | 27.7 |
| $q_3$ | 150-167 ppm | $\underline{C}_{ar}$O | 10.3 | 12.5 | 11.1 | 8.7 | 9.0 |
| $q_4$ | 140-150 ppm | $C_{ar}$C, $\underline{C}_{ar}$O (ipso-C in 1,3,5-trioxobenzenes) | 6.2 | 6.1 | 6.4 | 5.8 | 5.9 |
| $q_5$ | 120-140 ppm | $C_{ar}$C benzenes | 13.1 | 13.8 | 14.4 | 13.0 | 14.0 |
| $q_6$ | 108-120 ppm | $C_{ar}$C (ipso-C in 1,3,5-trioxobenzenes) | 7.2 | 9.7 | 8.0 | 6.1 | 7.6 |
| $q_7$ | 90-108 ppm | $\underline{C}_{ar}$C (ipso-C in 1,3,5-trioxobenzenes), $O_2\underline{C}_qC_2$ | 8.5 | 11.1 | 9.6 | 7.9 | 8.3 |
| $q_8$ | 66-90 ppm | $O\underline{C}_qC_3$ | 8.3 | 8.1 | 9.7 | 8.1 | 10.0 |
| $q_9$ | 20-66 ppm | $\underline{C}_qC_4$ | 15.7 | 8.3 | 11.2 | 15.8 | 10.3 |

$^{13}$C NMR-derived proportions of $C_{sp2}$-based and $C_{sp3}$-based quaternary carbon $C_q$ (that is, carbon not carrying any hydrogen according to NMR notation) in five DOM (SNATB), normalized to 100% total $^{13}$C NMR integral of $C_q$ units.

**Extended Data Table 6 | Chemical reactions in the synthesis and degradation of DOM molecules**

| item | reaction | molecular change | change of carbon atomic environments |
|------|----------|------------------|--------------------------------------|
| 1 | hydrogenation | $+H_2$ | $2\times Csp^2$ into $2\times Csp^3$ |
| 2 | dehydrogenation | $-H_2$ | $2\times Csp^3$ into $2\times Csp^2$ |
| 3 | hydration | $+H_2O$ | $2\times Csp^2$ into $2\times Csp^3$ |
| 4 | dehydration | $-H_2O$ | $2\times Csp^3$ into $2\times Csp^2$ |
| 5 | carboxylation | $+CO_2$ | insertion only, hybridization stays for all other carbon atoms |
| 6 | decarboxylation | $-CO_2$ | release only, hybridization stays for all other carbon atoms |
| 7 | methylation | $+CH_2$ | insertion only, hybridization stays for all other carbon atoms |
| 8 | demethylation | $-CH_2$ | release only, hybridization stays for all other carbon atoms |
| 9 | oxidation (CHO molecules) | $+O$ | insertion only, hybridization stays for all other carbon atoms |
| 10 | reduction (CHO molecules) | $-O$ | release only, hybridization stays for all other carbon atoms |
| 11 | carbonylation | $+CO$ | insertion only, hybridization stays for all other carbon atoms |
| 12 | decarbonylation | $-CO$ | release only, hybridization stays for all other carbon atoms |
| 13 | oxidative dearomatization ODA | molecular composition remains intact (tautomerization) | $1\times Csp^2$ into $1\times Csp^3$ |
| 14 | [4+2] cycloaddition following ODA | molecular mass doubles; elemental ratios remain intact | $4\times Csp^2$ into $4\times Csp^3$ |
| 15 | ODA / general cycloaddition, identical molecules $2\times A$ | molecular mass doubles from A to $2\times A$; elemental ratios remain intact | $5\times Csp^2$ into $5\times Csp^3$ |
| 16 | ODA / general cycloaddition, distinct molecules A and B | molecular mass increases to A+B; elemental ratios remain intact | $5\times Csp^2$ into $5\times Csp^3$ |

Chemical reactions in the synthesis and degradation of DOM molecules and how these affect carbon chemical environments. Reactions 1–12 have been previously described in organic matter research. ODA activates oxygenated aromatic rings in DOM, such as lignins and tannins, making the generated cyclohexadienones subject to consecutive cycloaddition reactions, of which the most prominent is of the [4+2] type, that is, Diels–Alder cycloaddition. All reactions 1–16 can be of biotic or/and abiotic origin.