## [Peer Review File · Nature]

Manuscript Title: Dearomatization drives complexity generation in freshwater organic matter

Redactions – Third Party Material

Reviewer Comments & Author Rebuttals

Reviewer Reports on the Initial Version:

Referees' comments:

Referee #1 (Remarks to the Author):

The key results presented are a carbon-based analysis of DOM, specifically focussing on the level of hydrogenation of carbons (C_n where $n=0-3$), assessed by NMR spectroscopy. From these data, the authors extrapolate the origin of DOM to be primarily arising from polyphenol oxidation chemistry (and subsequent diversity-inducing rearrangements, ring-openings and substitutions).

The originality of this work is primarily in the identification of the C_n diversity of DOM by NMR (Fig 2), and the higher structural-level information this provides. This data is indeed novel to the best of my knowledge, and provides interesting insight into the nature of DOM, beyond the usual mass spectrometric analyses. However, I have significant concerns over the quantitative value of these NMR data, and the experimental details provided do not allow me to assess how reliably quantitative these NMR data would be (for example, the ^{13}C spectra used 19s relaxation delays, and the DEPT used 3s, which are both very short delay periods for quantitative acquisitions. Similarly the initial NMR excitation (flip angle) is not given, which impacts how long the relaxation period needs to be. Also, the nature of the decoupling and/or inverse-gating (which affects quantitation via NOE build-up) is not mentioned.

The extrapolation of the C_n information in Figure 1 to the subsequent diversity-inducing chemistry (Fig 2) is very unclear. There is no information presented in Fig 1 which directly supports Fig 2 - instead Fig 2 seems to be a proposed set of reaction chemistries which might give rise to C_n diversity, but there is no evidence that supports this other than the highly quaternary nature of DOM, instead the authors simply state that these pathways exist and are consistent with the NMR data - but that would be true of many other pathways as well.

Overall, I am not convinced that the Conclusions drawn can be supported by the data measured, and that the significance of this work lies primarily in the more structural-level data provided by NMR, but this is not enough to draw the subsequent polyphenol-derived diversity conclusions (it seems obvious that this is route to be honest given the input compounds to DOM, but the data doesn't really exclude any other pathways)

Referee #2 (Remarks to the Author):

Advances in NMR spectroscopy and mass spectrometry have revealed an amazing molecular diversity and complexity in freshwater and marine DOM. The diversity of molecules in DOM greatly exceeds the known diversity of common biomolecules, so it has been assumed that biotic and

abiotic processes increase the molecular diversity and complexity of DOM over time. Microbial degradation processes and the activity of promiscuous enzymes appear to play a major role in shaping the molecular diversity and complexity of marine DOM (Benner and Amon 2015), but the specific mechanisms driving DOM complexity are unclear. The study by Li et al. provides exciting and novel insights about how oxidative dearomatization (ODA) could play an important role in shaping the molecular diversity and complexity of freshwater DOM.

The authors present substantial evidence indicating ODA plays a role in shaping the molecular diversity and complexity of the DOM in the sampled rivers and lakes. These freshwater systems are rich in plant-derived polyphenols that are susceptible to ODA. In this study, ¹³C NMR characterization of freshwater DOM from 4 rivers in the Amazon and 2 boreal lakes revealed great complexity and molecular diversity. The DOM in these rivers and lakes is very rich in quaternary-C (>50% of OC), most of which is carboxylic groups. The sum of quaternary-C and methine-C is >80% of the OC. These characteristics are indicative of carboxyl-rich alicyclic molecules (CRAM), a diverse and abundant form of DOM in fresh and marine waters. Overall, these observations indicate structural rearrangement and molecular diversification of DOM. The authors propose these molecular transformations can be explained by ODA of polyphenols, and they present compelling information (e.g. Fig. 3) to support ODA as a mechanism for the structural rearrangement and diversification of DOM.

How important are ODA reactions in the formation of molecular diversity and CRAM in aquatic systems that have minimal inputs of plant-derived polyphenols? The global ocean is the largest reservoir of DOM, and the concentrations of plant-derived polyphenols in the ocean are extremely low. Ocean DOM, like freshwater DOM, has high molecular diversity and complexity and an abundance of CRAM. Marine microbes grown in the dark with simple, non-aromatic substrates (e.g. glucose) rapidly produce DOM that is structurally complex and has a diversity of carboxyl-rich alicyclic molecules (Lechtenfeld et al. 2015). Microbes are not a source of polyphenols, so it seems unlikely that ODA is the mechanism for generating carboxyl-rich alicyclic molecules in these experiments. It appears microbially-mediated processes also generate molecular diversity and complexity (e.g. CRAM) in DOM. Can ODA and microbially-mediated DOM transformations be chemically distinguished? Experiments with specific polyphenols or glucose and freshwater microbial assemblages would be useful for providing insights about these processes. The authors should consider whether ODA is the only driver of complexity generation in freshwater DOM.

Specific Comments

Line 23-24: global soil OC is complex and is a larger reservoir than DOC

Line 311: revise "andedict"

Line 544: the Tapajos River sample should be T-DOM rather than B-DOM

Provide the names of the river and lake samples in the figure and table legends.

Reviewer – Ron Benner

**Referee #1 (Remarks to the Author):**

The key results presented are a carbon-based analysis of DOM, specifically focusing on the level
of hydrogenation of carbons (C_n where $n=0-3$), assessed by NMR spectroscopy.

Yes, the reviewer is correct in identifying the ^{13}C NMR subspectra of nine C_q atom environments
according to NMR notation. To prepare for some of our clarifications below we would like to highlight
the significance of multiplicity-edited ^{13}C NMR spectra. Here, C_q are carbon atoms carrying no hydrogen;
these decompose into seven familiar C_q units with sp^2 -hybridized carbon (Extended data Fig. 3 in
publication), and two C_q units with sp^3 -hybridized carbon, i.e. C_qC_4 and OC_qC_3 units, four methine CH
atom environments ($C_{ar}H$, O_2CH , OCH and CCH units), two methylene CH_2 atomic environments
(OCH_2 and CCH_2 units) and two methyl CH_3 atom environments (OCH_3 and CCH_3 units). This defines
17 groups of carbon atomic environments in DOM. This additional resolution greatly enhances the
opportunities for interpreting the ^{13}C NMR spectra of dissolved organic matter (DOM) and led to the
observation of OC_qC_3 units in tropical and boreal DOM with high abundance. OC_qC_3 units play a critical
role in the processing of DOM molecules and are products when oxygenated benzene derivatives, such as
lignin and tannin degradation products, are subjected to oxidative dearomatization (ODA).

Another major outcome of this spectral editing of ^{13}C NMR spectra was the surprising finding that
tropical river (Amazon) and boreal lake (Sweden) DOM comprised up to ~65% of C_{sp^2} - and C_{sp^3} -based
quaternary carbon, and showed >80% of the sum of quaternary and methine carbon. This implies high
degrees of compaction of DOM molecules on average, in which most carbon atoms are bound to several
other carbon atoms.

From these data, the authors extrapolate the origin of DOM to be primarily arising from
polyphenol oxidation chemistry (and subsequent diversity-inducing rearrangements, ring-
openings and substitutions).

We appreciate this comment highlighting a need for clarification in our text. We attribute a unique role to
the abundant OC_qC_3 carbon atomic units in the DOM, revealing oxidative dearomatization (ODA) of the
abundant lignin and tannin degradation products to be the most probable major DOM shaping
mechanism. ODA resides on a tautomerization of phenols to cyclohexadienones and therefore is NOT a
polyphenol oxidation chemistry. Energy-rich cyclohexadienones are subject to a diverse array of
consecutive and parallel chemical transformations, in which cycloadditions play a relevant role.
Cycloadditions thoroughly change the molecular skeletons of participants, with significant interchange of
C_{sp^2} - and C_{sp^3} -hybridized carbon atoms. Classical polyphenol oxidation chemistry also does NOT apply in
these reactions. We have now tried to clarify this. Sorry for not realizing this unclarity from the
beginning.

The originality of this work is primarily in the identification of the C_n diversity of DOM by NMR
(Fig 1), and the higher structural-level information this provides. This data is indeed novel to the
best of my knowledge, and provides interesting insight into the nature of DOM, beyond the
usual mass spectrometric analyses.

Thanks for this notion. While this extent of multiplicity editing for ^{13}C NMR spectra of DOM clearly
reaches beyond current literature, two major novelties stand out even more, i.e. the identification of nine
abundant chemical environments of C_q in DOM, and the previously not described quaternary aliphatic
carbon environments C_qC_4 , OC_qC_3 and the minor $O_2C_qC_2$ units. Here, the OC_qC_3 units carry unique
significance because those convincingly corroborate the central role of ODA in the processing of DOM
molecules. OC_qC_3 units are virtually absent in all common biomolecules but very abundant in tropical
and boreal DOM. ODA of lignin and tannin derived polyphenols directly produces OC_qC_3 units.

However, I have significant concerns over the quantitative value of these NMR data, and the
experimental details provided do not allow me to assess how reliably quantitative these NMR
data would be (for example, the ^{13}C spectra used 19s relaxation delays, and the DEPT used 3s,
which are both very short delay periods for quantitative acquisitions. Similarly the initial NMR
excitation (flip angle) is not given, which impacts how long the relaxation period needs to be.
Also, the nature of the decoupling and/or inverse-gating (which affects quantitation via NOE
build-up) is not mentioned.

Thanks a lot for highlighting this lack of clarity. These are important aspects and therefore we provide a
detailed response below on quantification of DOM substructures by means of ^{13}C NMR spectroscopy
addressing the “*significant concerns over the quantitative nature of these NMR data*”, and “*the initial*
*NMR excitation (flip angle) is not given*”.

In the experimental section, we have provided 90-degree pulses ($\sim 10\ \mu\text{s}$) and acquired DEPT-45, 90, 135,
QUAT, and spin-echo single pulse ^{13}C NMR experiments, which require 90-degree pulses (other
combinations like 38/90/142-deg pulses carry no real advantages in mixtures of molecules and render
individual DEPT subspectra useless for dedicated analysis). We have used inverse-gated decoupling
[which provides quantification unbiased by nuclear Overhauser effect (nOe)]. However, we have not
mentioned the use of inverse-gated decoupling specifically in the original manuscript. We now have
corrected this omission at Line 609 in supplementary material.

In order to correctly address longitudinal relaxation time for ^{13}C nuclei in our DOM molecules, the delay
time (d1+aq; the sum of relaxation delay and acquisition time) used for single pulse and QUAT ^{13}C NMR
spectra was 20 s and thus “exceedingly” long compared to all published solution-state ^{13}C NMR spectra
of NOM (Lam et al., 2007, Thorn et al., 1989). NOM is a complex mixture of polydisperse and
molecularly heterogeneous molecules with high content of carboxylic groups and highly oxygenated
aromatic molecules. Solution state ^{13}C NMR spectroscopy offers atomic mobility within DOM molecules
with accelerated T_1 relaxation (longitudinal relaxation), which is not available to solid state ^{13}C NMR.
Nevertheless, recycle delays (d1+aq) for quantitative direct polarization (DP) ^{13}C NMR spectra of many
DOM were 15 s (Dria et al., 2002), and could reach 50-200 s in exceptional cases (Mao et al., 2004 and
2017). Such long T_1 values refer solely to solid-state ^{13}C NMR spectra of DOM. Abundant carboxylic
groups (and to lesser extent, hydroxyl groups as well) mediate intermolecular interactions in polydisperse
mixture of DOM molecules (e.g. via hydrogen bonds and through polarization), impeding mobility of
individual atoms. Furthermore, oxygenated aromatic molecules are credible carriers of organic radicals in
NOM, and are observed by Electron Spin Resonance (ESR; Scott et al., 1998); the electron magnetic
moment is 658.2 times larger than the proton magnetic moment, and accelerates NMR relaxation across
appreciable distances in molecules (intramolecular) and beyond (intermolecular).

Hashim Farooq et al., a member of the Andre Simpson group, Department of Chemistry, University of
Toronto, a leading research group concerning solution-state NMR spectroscopy of NOM, published
expedient T_1 measurements of dissolved organic matter (Rapid parameter optimization of low signal-to-
noise samples in NMR spectroscopy using rapid CPMG pulsing during acquisition: application to recycle
delays, *Magnetic Resonance in Chemistry*, **51** (2013) 129-135). Here, they found T_1 of 1.28 s, leading to
99% relaxation at 6.4 s ($5 * T_1$). Other groups have used recycle delays of 8-12 s for solution state ^{13}C
NMR of various NOM (Lam et al., 2007; Thorn et al., 1989); all our previous own contributions used
either 15 or 20 s (Hertkorn et al., 2002, 2006, 2007, 2013, 2016; Einsiedl et al., 2007; Zhang et al., 2014).
All these publications of solution state ^{13}C NMR of NOM were consistent and give confidence that our
delay of 20 s is very suitable for quantification. Therefore, we have not specifically addressed this general
knowledge in our original contribution. In fact, we have used screening experiments also for SNATB-
DOM with delays of 4, 6 and 8 s and found nearly identical distribution of ^{13}C NMR resonance integrals;
however, we did not mention those in the text. We are grateful that the review brought this question to
our attention so we could clarify these aspects in the revised text, and we agree with the review point to

not take important methodological aspects for granted. It is easy to miss such things when becoming
“home blind” so thanks for attentive review reading.

For specifically addressing reviewer #1’s concerns, we have repeated acquisition of DEPT-45 and single-
pulse (spin-echo) ^{13}C NMR spectra for the consolidated sample N-DOM (Rio Negro solid phase extracted
(SPE)-DOM) for variable combinations of relaxation delay (d1) and acquisition time (aq) as shown in
Fig. R1. T_1 relaxation in ^{13}C DEPT NMR spectra is governed by relaxation of ^1H nuclei, whereas
relaxation in single pulse ^{13}C NMR spectra is governed by ^{13}C nuclei; hence, (d1+aq) adequate for
quantification is shorter for ^{13}C DEPT-45 NMR spectra than for single pulse ^{13}C NMR spectra.

Fig. R1. Area-normalized DEPT-45 ^{13}C NMR spectra (125 MHz, $^{12}\text{CD}_3\text{OD}$; ns = 9246) of consolidated sample N-DOM (Rio Negro SPE-DOM), with (A) stacked plot, (B) direct overlay, and (C) direct overlay of computed ^{13}C DEPT-45 spectra with 1 ppm bucket resolution. Relaxation delay (d1) and acquisition time (aq) were (d1/aq [sec]) (a) 0.5/0.5; (b) 0.5/1; (c) 1/1; (d) 1.5/1; (e) 2/1; (f) 3/1; (g) 4/1. Near identity of ^{13}C NMR section integrals was attained already at (d1+aq) \approx 1.5 sec.

Fig. R2. Area-normalized single-pulse spin-echo ^{13}C NMR spectra (125 MHz, $^{12}\text{CD}_3\text{OD}$; ns = 7168) of consolidated sample N-DOM (Rio Negro SPE-DOM), with (A) stacked plot, (B) direct overlay, and (C) direct overlay of computed single-pulse spin-echo ^{13}C NMR spectra with 1 ppm bucket resolution; virtual identity of ^{13}C NMR spectra was already attained at $(d1+aq) \approx 4$ sec; OC_qC_3 , O_2CCH and $\text{O}_2\text{C}_q\text{C}_2$ units showed longer T_1 as seen for ^{13}C NMR acquired at $(d1+aq) = 2$ sec. This indicates an appreciable distance between those carbon atoms and closest protons on average, and corroborates the deep embedding of these structural units in DOM molecules, distant from the hydrogen-rich molecular surface. We have included a more extended appreciation in the NMR experimental section.

The extrapolation of the C_n information in Figure 1 to the subsequent diversity-inducing
chemistry (Fig 2) is very unclear. There is no information presented in Fig 1 which directly
supports Fig 2 - instead Fig 2 seems to be a proposed set of reaction chemistries which might
give rise to C_n diversity, but there is no evidence that supports this other than the highly
quaternary nature of DOM, instead the authors simply state that these pathways exist and are
consistent with the NMR data - but that would be true of many other pathways as well.
Again many thanks for highlighting unclear aspects of our study. This concern is now addressed in two
steps as follows (*here and next paragraph*).

Aliphatic quaternary carbon units C_qC₄ and OC_qC₃ convey elaborate three dimensional shapes to DOM
molecules. The high abundance of C_q in all DOM is of remarkable structural relevance, but the presence
of abundant tetrahedral core carbon units C_qC₄ and OC_qC₃ in tropical riverine and boreal lake DOM is an
independent and more stringent structural restriction compared with Csp²-related unsaturation. It is
associated with aliphatic hydrogen-deficient molecules.

(Aliphatic) sp³-hybridized carbon atoms show tetrahedral geometry and are connected with four other
atoms whereas sp²-hybridized carbon atoms show trigonal planar geometry and are connected with three
other atoms. Carbon atoms do not terminate chemical bonds in common CHO-molecules, but e.g. -H, -
CH₃, -OH and -COOH do terminate chemical bonds and therefore reside rather at the surface of
molecules. Hence, C_qC₄ and OC_qC₃ structural units are the ultimate carriers of aliphatic branching,
deeply embedded in molecules by definition; they convey elaborate three-dimensional, natural product-
like structures to many DOM molecules as well as structural recalcitrance, because these aliphatic C_q
atoms are connected to several other carbon atoms and not readily removed from DOM molecules.

Overall, I am not convinced that the Conclusions drawn can be supported by the data measured,
and that the significance of this work lies primarily in the more structural-level data provided by
NMR, but this is not enough to draw the subsequent polyphenol-derived diversity conclusions
(it seems obvious that this is route to be honest given the input compounds to DOM, but the
data doesn't really exclude any other pathways)

We now try to better explain our findings. Our key point is that OC_qC₃ carbon atomic units result from
oxidative dearomatization (ODA) of abundant lignin and tannin derived polyphenols in DOM.

There is a fundamental distinction between C_qC₄ and OC_qC₃ units in terms of backtracking their
formation. Many different reactions might eventually produce C_qC₄ atomic units, comprising e.g.
rearrangement reactions and also “selective preservation” of sterically protected C_q carbon atoms in
precursor molecules of NOM. Therefore, presence of C_qC₄ units in DOM may not enable specific
conclusions about singular reactions leading to their formation. This agrees with the comment of this
reviewer.

This situation is however very different for OC_qC₃ atomic units, which are exceedingly rare in all
common biomolecules and primary metabolites, but are readily formed by ODA of oxygenated benzene
derivatives that are very abundant in tropical and boreal DOM. Therefore, ODA of lignin- and tannin-
derived polyphenols is the most direct reaction pathway to abundant OC_qC₃ units in DOM.

We were quite puzzled about the high abundance of OC_qC₃ atomic units both tropical and boreal
freshwater DOM, and looked for plausible explanation for several weeks. There is no conceivable
pathway leading from joint processing of the major biomolecules peptides, lipids, and carbohydrates,
which lack OC_qC₃ units but take center stage in common literature of DOM processing, to synthesize
OC_qC₃ units in relevant abundance. Apart from selective preservation of OC_qC₃ atomic units, this applies
also for all other classes of conceivable precursor molecules of DOM. Selective preservation of OC_qC₃
atomic units fundamentally deviates from their synthesis during processing of DOM. After extensive
literature search, Fig. 1 in Chandra and Patel, 2002 provided a display of 24 molecules which all

contained OC_qC₃ units (Fig. R3A). This was our first lead, and follow-up search readily produced
evidence for straightforward relationships between the occurrence of OC_qC₃ atomic units in DOM and
ODA of lignin- and tannin-derived polyphenols. Fig. R3 shows three examples out of several dozens of
recent publications referring to the use of ODA in contemporary organic synthesis. Accordingly, given
the current knowledge that we were able to find in the English scientific literature, ODA represents the
only realistic explanation for the high relative abundance of OC_qC₃ atomic units in DOM molecules, and
such high prevalence of ODA also provides a remarkable straightforward explanation for the observed
DOM complexity in nature. We have now tried to clarify this in the revised manuscript.
We have also added the following sentences to the manuscript to accommodate two alternative pathways
to OC_qC₃ units. However, both pathways cause incremental, additive molecular transformations
(Extended Data Fig. 5) but are not capable of generating topological complexity from structurally simple
precursors as realized by ODA.

Two other environmental synthesis pathways to produce OC_qC₃ carbon units are known but
appear to be of minor relevance compared to ODA. One is selective preservation of OC_qC₃ units
in precursor (bio)molecules like oxygenated terpenoids^{DeVault_2022, Wu_2006}. Another is the
unselective attack of energy-rich hydroxyl radicals on DOM molecules^{Wennberg_2018}. Hydroxylation
also may create OC_qC₃ carbon units from suitable aliphatic precursors^{Li_2021}. However, both
pathways cause incremental, additive molecular transformations (Extended Data Fig. 5a) but are
not capable of generating topological complexity from structurally simple precursors as realized
by ODA (Extended Data Fig. 5b; ^{Huck_2022}). The rather diffuse input of OC_qC₃ units from highly
diverse molecules into the ecosphere caused by these reactions is very probably not competitive
with ODA in the molecular transformation of boreal and tropical DOM, in which up to 50% of
carbon can be related with structural features susceptible to ODA either as educts (polyphenols)
or products (OC_qC₃ units) according to ¹³C NMR spectra.

Our conclusions referring to the key role of ODA in creating structural complexity in DOM molecules is
based on the collective related knowledge we could access. If the reviewers are aware of other
alternatives that are realistic for the high observed abundance of OC_qC₃ atomic units we would be happy
for specific mechanistic information so we can include them in the analysis and modify our conclusions.
We were first surprised and skeptical ourselves but conclusions based on the best available evidence need
to be accepted until specific contradicting evidence can be presented.

{REDACTED}

**Referee #2 (Remarks to the Author):**

Advances in NMR spectroscopy and mass spectrometry have revealed an amazing molecular
diversity and complexity in freshwater and marine DOM. The diversity of molecules in DOM
greatly exceeds the known diversity of common biomolecules, so it has been assumed that
biotic and abiotic processes increase the molecular diversity and complexity of DOM over time.
Microbial degradation processes and the activity of promiscuous enzymes appear to play a
major role in shaping the molecular diversity and complexity of marine DOM (Benner and Amon
2015), but the specific mechanisms driving DOM complexity are unclear. The study by Li et al.
provides exciting and novel insights about how oxidative dearomatization (ODA) could play an
important role in shaping the molecular diversity and complexity of freshwater DOM.

*We thank reviewer #2 for all above, correct comments. What has been said here for marine DOM applies
conceptually to freshwater DOM as well. However, we clearly state here, that the molecular structures of
terrestrial/freshwater and marine DOM are expected to differ in many aspects, and this may also apply to
their respective precursor molecules and formation histories. It is not unlikely that our results apply to
marine conditions but we want to be careful to constrain our conclusions to the type of organic matter we
have studied.*

The authors present substantial evidence indicating ODA plays a role in shaping the molecular
diversity and complexity of the DOM in the sampled rivers and lakes. These freshwater systems
are rich in plant-derived polyphenols that are susceptible to ODA.

*Yes, and although ODA may be of global importance for DOM complexity we prefer to be careful and
constrain our conclusions to the freshwater DOM that we studied.*

In this study, ¹³C NMR characterization of freshwater DOM from 4 rivers in the Amazon and 2
boreal lakes revealed great complexity and molecular diversity. The DOM in these rivers and
lakes is very rich in quaternary-C (>50% of OC), most of which is carboxylic groups. The sum of
quaternary-C and methine-C is >80% of the OC. These characteristics are indicative of carboxyl-
rich alicyclic molecules (CRAM), a diverse and abundant form of DOM in fresh and marine
waters. Overall, these observations indicate structural rearrangement and molecular
diversification of DOM. The authors propose these molecular transformations can be explained
by ODA of polyphenols, and they present compelling information (e.g. Fig. 2) to support ODA as
a mechanism for the structural rearrangement and diversification of DOM.

*We thank reviewer #2 for all above, correct comments. This comment is an extension of our study (of
tropical and boreal freshwater DOM). We too suspect that ODA might apply to e.g. marine DOM of
entirely different origin and processing history, but without corresponding data for marine DOM we do
not at this point make strong statements about it in the manuscript.*

*As stated below, we propose complementary roles for ODA and chemistry of carboxylic acids in our
manuscripts. ODA offers singular capacity to generate structural complexity from polyphenols which are
abundant in freshwater DOM. Consecutive oxidation of ODA products fosters production of CRAM, and
carboxylic acids show own chemistry, relevant in further compaction of DOM molecules. The manuscript
has been extensively reworded to better describe the complementary roles of ODA and COOH chemistry
in the processing of DOM.*

How important are ODA reactions in the formation of molecular diversity and CRAM in aquatic
systems that have minimal inputs of plant-derived polyphenols? The global ocean is the largest

reservoir of DOM, and the concentrations of plant-derived polyphenols in the ocean are
extremely low.

Thanks for this good question. While “common, land based” plant derived polyphenols like lignins and
tannins obviously show very low concentration in the open ocean, phlorotannins (Powers et al., 2019)
that are synthesized by several marine organisms may show a higher relative abundance than remnants of
processed land based polyphenols. The 1,3,5-symmetry of benzene rings in phlorotannins will initiate
specific reactivity for ODA, distinct from other polyphenols. While the concentration of marine
polyphenols is very low on average, ODA of those is conceivable, and ODA products will be oxygenated
aliphatic molecules that are distinct from precursor polyphenols. Unfortunately we do not have data for
marine DOM, but this discussion leads to an important hypothesis on what to expect for marine DOM.
Ocean DOM, like freshwater DOM, has high molecular diversity and complexity and an
abundance of CRAM.

This is correct. Marine and terrestrial/freshwater DOM derive from largely distinct pools of source
materials. While early stages of marine DOM primarily originate from primary production and associated
food webs and photochemical processing in near surface oceans, terrestrial/freshwater DOM incorporates
sizable shares of biogeochemical polyphenols like lignin and tannin degradation products, which become
quickly degraded or precipitate and sediment in coastal seas following riverine transport from land to
sea.

The detection of such fundamental distinction of chemical structures of marine and freshwater DOM
molecules is masked by the very massive projection in even the highest resolving methods of organic
structural spectroscopy: mass peaks depict defined molecular compositions of many (thousands of)
isomeric molecules, and NMR spectra depict millions of atomic environments at limited ranges of
chemical shifts δ . This massive projection produces a superficial congruence of “simple” mass and NMR
spectra for marine and freshwater/marine DOM by necessity on visual inspection. However,
mathematical data treatment reveals significant distinction between freshwater and marine DOM like e.g.
distinct CHNO and CHOS molecular compositions (FTMS) and much higher relative abundance of
aromatic units in terrestrial/riverine NOM (NMR).

Marine microbes grown in the dark with simple, non-aromatic substrates (e.g. glucose) rapidly
produce DOM that is structurally complex and has a diversity of carboxyl-rich alicyclic molecules
(Lechtenfeld et al. 2015).

This is a very interesting comment. From our current understanding, the type of freshwater DOM that we
studied are dominated by plant derived lignin and tannin degradation products, but although microbial
DOM is likely a minor share of the freshwater DOM studied, there may be other cases in experimental
situations that would be interesting to study. As noted in the comments by and responses to Reviewer 1
some CRAM diversity can be created without ODA, but in any case with high abundance of OC_qC_3 units
ODA becomes a main candidate for generating DOM complexity.

Microbes are not a significant source of polyphenols in terms of abundance, although many organisms
synthesize phenolic natural products as secondary metabolites of (very) minor abundance. Our
contribution has investigated tropical freshwater DOM from the Amazon River and Swedish boreal lakes
for which high proportions of terrestrial derived lignin and tannin degradation products are well
established in the literature, and again in ^{13}C NMR spectra in our contribution. Microbes will be most
probably major contributors to process polyphenols under various environments conditions. Here, ODA
is probably a key reaction of microbial processing of polyphenols in freshwater environments which has
been overlooked entirely so far by the scientific community in the field of organic matter. In comparison,
ODA has emerged as one of the key drivers to generate complex organic molecules by the community of
synthetic organic chemistry in recent years.

**ODA and carboxylic acid chemistry carry complementary roles in the processing of DOM:**

ODA acts as a primary generator of complexity in DOM molecular structures, and *subsequent* carboxylic
acid chemistry acts as a critical modifier of NOM molecules during further processing. Both items have
not been described in literature with distinction.

ODA fundamentally generates structural complexity of DOM molecules in a single-step cascade reaction
(Fig. 2), and we propose carboxylation of ODA products as a straightforward pathway leading to CRAM.
The key role of ODA in DOM processing is the first key novelty in our manuscript. However, ODA does
NOT synthesize carboxylic acids (it may rearrange suitable precursor carboxylic acids as a minor reaction
throughout DOM processing).

Carboxylic groups are the defining feature of carboxyl-rich alicyclic molecules (CRAM) that are ubiquitous
in DOM across water systems (Hertkorn et al., 2006; 576 citations on Sept. 22, 2022). Fused alicyclic rings
showed better conformity with NMR properties in 2006 (Fig. 13; GCA_2006) than other structures; no
further extrapolations were done in 2006.

COOH is a highly reactive attachment C_q unit, whereas all other C_q atoms in DOM molecules are
connected to two or more carbon atoms. CRAM observance in ¹³C NMR spectra implies the co-
occurrence of (aliphatic and aromatic) carboxylic acids and alicyclic rings in DOM “on average”.
However, the high abundance of both structural units in DOM, and the considerable size of DOM
molecules (Li et al., 2023), infers presence of both substructures in a majority of DOM molecules. The
positioning of COOH towards the surface of DOM molecules conveys independent reactivity, including
decarboxylative functionalization and carboxylation through complementary neutral, ionic, and radical
pathways. Microbial and abiotic oxidation of DOM uses molecular oxygen and/or reactive oxygen
species to generate carboxyl groups, an efficient processing step of DOM in oxic surface waters.

COOH chemistry is a critical modifier in the structural evolution of DOM towards more compact
molecules during environmental processing, which increases the average number of chemical bonds
between constituent atoms in DOM molecules, and the proportions of quaternary and methine carbon
units at the expense of methylene and methyl units (Fig. 2d). For instance, intermediates produced by
decarboxylation carry intrinsic energy fostering structural rearrangements. In particular, free radicals have
distinct reactivity, and skeletal rearrangements toward higher compaction are supported by the higher
stability of sterically crowded radical positions, opposite to common chemistry in which increasing steric
demands (e.g. entry of new substituents to preexisting molecular atomic environments) are difficult to
attain. Intramolecular reactions with participation of abundant carboxyl and hydroxyl groups contribute to
other compaction of DOM molecules, by e.g. forming anhydrides (two COOH groups), lactones (COOH
and OH units), and ethers (two OH units); (Extended Data Fig. 5).

¹³C NMR spectra of freshwater DOM show high proportions of carboxylic acids, in the range of 15-17%,
a value higher than found in almost all metabolites and natural products. While carboxylic groups are the
most abundant group of C_q units, they overall contribute ~one quarter of all C_q atomic units (Extended
Data Table 5 in original manuscript). Most of C_q in DOM comprise oxygenated aromatic units and
skeleton-forming Csp³-based aliphatic units OC_qC₃ (with minor admixture of O₂C_qC₂) and C_qC₄ atomic
units.

so it seems unlikely that ODA is the mechanism for generating carboxyl-rich alicyclic molecules
in these experiments.

The distinction of ODA, which does NOT create CRAM (except in case of rearranging suitable precursor
carboxylic acids), and the COOH chemistry which contributes INDEPENDENTLY to the processing of
DOM is a key point of our publication. ODA and COOH chemistry carry complementary roles in the
processing of DOM.

It appears microbially-mediated processes also generate molecular diversity and complexity
(e.g. CRAM) in DOM. Can ODA and microbially-mediated DOM transformations be chemically
distinguished? Experiments with specific polyphenols or glucose and freshwater microbial
assemblages would be useful for providing insights into these processes. The authors should
consider whether ODA is the only driver of complexity generation in freshwater DOM.

ODA has already been shown to operate by a variety of microbial and abiotic mechanisms in the lab and
our work illustrates that this is relevant also in nature. Owing to mixture complexity in the case of DOM,
both types of ODA (microbial vs abiotic) cannot be differentiated at present. We very much appreciate
this critical comment and have, as the reviewer suggests, carefully considered all different drivers of
DOM diversity that we could find in the literature or imagine, and until any explicit contradicting
evidence is provided we cannot reach any other conclusion than the following: ODA is certainly not the
only possible driver of complexity generation in freshwater DOM (Extended Data Fig. 5), but (a) given
best available evidence, ODA is a major diversity driver where there is a high abundance of OC₄C₃
atomic units, and (b) such high abundance of OC₄C₃ atomic units were found in all analyzed freshwater
DOM from both the Amazon and the boreal regions indicating global consistency in the importance of
ODA in shaping DOM diversity (see also above responses).

Classical incremental processing of DOM molecules by addition and release of small substructures like
e.g. ± H₂, CH₂, O, CO, CO₂ will operate alongside ODA (Extended Data Fig. 5). However, these
reactions rather modify substituents in quite sizable molecules and do not have the potential to
fundamentally change molecular structures of the carbon skeleton as effected by ODA. These issues have
been clarified in the manuscript.

Specific Comments

Line 23-24: global soil OC is complex and is a larger reservoir than DOC

Line 311: revise "andedict"

Line 544: the Tapajos River sample should be T-DOM rather than B-DOM

Provide the names of the river and lake samples in the figure and table legends.

Thank you for pointing out these errors; we have corrected those.

We sincerely thank the reviewers for their insightful comments which helped us to improve our wording
in the manuscript. This substantially improved the description of the key novel aspects of DOM
processing, i.e. ODA as main complexity generator, and COOH chemistry as a critical modifier of DOM
molecules.

**Norbert Hertkorn, on behalf of all coauthors**

REFERENCES

- Chandra, G., and Patel S., Molecular complexity from Aromatics: Recent Advances in the Chemistry of para-Quinol
and Masked para-Quinone Monoketal, *Chemistry Select* **5**, 12885-12909 (2020).
- Dria, K.J., Sachleben, J.R., Hatcher, P.G., Solid-State Carbon-13 Nuclear Magnetic Resonance of Humic Acids at
High Magnetic Field Strengths, *J. Environ. Qual.*, **31**, 393-401 (2002).
- Einsiedl, F., Hertkorn, N., Wolf, M., Frommberger, M., Schmitt-Kopplin, P., Koch, B. P. Rapid biotic molecular
transformation of fulvic acids in a karst aquifer, *Geochim. Cosmochim. Acta*, **71**, 5474-5482 (2007).
- Farooq, H., et al., Rapid parameter optimization of low signal-to-noise samples in NMR spectroscopy using rapid
CPMG pulsing during acquisition: application to recycle delays, *Magn. Reson. Chem.*, **51**, 129-135 (2013).
- Hertkorn, N., Permin, A., Perminova, I., Kovalevskii, D., Petrosian, V.M., Kettrup A., Comparative Analysis of
Partial Structures of a Peat Humic and Fulvic Acid Using One- and Two-Dimensional Nuclear Magnetic
Resonance Spectroscopy, *J. Environ. Qual.* **31** (2002) 375-387.
- Hertkorn, N., Benner, R., Schmitt-Kopplin, P., Kaiser K., Kettrup A., Hedges, I. J. Characterization of a major
refractory component of marine organic matter, *Geochim. Cosmochim. Acta*, **70**, 2990-3010 (2006).
- Hertkorn, N., Ruecker, C., Meringer, M., Gugisch, R., Frommberger, M., Perdue, E. M., Witt, M., Ph. Schmitt-
Kopplin P., High-precision frequency measurements: indispensable tools at the core of the molecular-level
analysis of complex systems, *Anal. Bioanal. Chem.*, **389**, 1311-1327 (2007).
- Hertkorn, N., Harir, M., Koch, B. P., Michalke, B., Schmitt-Kopplin, P. High field NMR Spectroscopy and FTICR
Mass Spectrometry: Powerful Discovery Tools for the Molecular Level Characterization of Marine Dissolved
Organic Matter, *Biogeosciences* **10**, 1583-1624 (2013).
- Hertkorn, N., Cawley, K., Harir, M., Schmitt-Kopplin, P., Jaffe, R., Molecular characterization of dissolved organic
matter from subtropical wetlands: a comparative study through the analysis of optical properties, NMR and
FTICR/MS, *Biogeosciences*, **13**, 2257-2277 (2016).
- Huck, C. J., Boyko, Y. D., Sarlah, D. (2022) Dearomative logic in natural product total synthesis, *Nat. Prod. Rep.*,
**39**, 2231-2291.
- Lam, B., Baer, A., Alace, M., Lefebvre, B., Moser, A., Williams, A., Simpson, A. J. Major Structural Components
in Freshwater Dissolved Organic Matter, *Environ. Sci. Technol.*, **41**, 8240-8247 (2007).
- Lechtenfeld, O. J., Hertkorn, N., Shen, Y., Witt, M., Benner, R. Marine Sequestration of Carbon in Bacterial
Exometabolites, *Nature Communications*, **6**, 6711 (2015).
- Li, S., et al., Comprehensive assessment of dissolved organic matter processing in the Amazon River and its major
tributaries revealed by positive and negative electrospray mass spectrometry and NMR spectroscopy, *Sci. Total*
*Environ.*, **857**, (2023), 159620.
- Mao, J., Cao, X., Olk, D.C., Chu, W., Schmidt-Rohr, K., Advanced solid-state NMR spectroscopy of natural organic
matter, *Prog. Nucl. Magn. Reson. Spec.*, **100**, 17-51 (2017).
- Mao, J., Schmidt-Rohr, K., Accurate Quantification of Aromaticity and Nonprotonated Aromatic Carbon Fraction in
Natural Organic Matter by ¹³C Solid-State Nuclear Magnetic Resonance, *Environ. Sci. Technol.*, **38**, 2680-2684
(2004).
- Powers, L., Hertkorn, N., McDonald, N., Schmitt-Kopplin, P., Del Vecchio, R., Blough, N., Gonsior, M. Sargassum
sp. act as a Large Regional Source of Dissolved Organic Carbon and Polyphenols to the Ocean, *Global*
*Biogeochemical Cycles* **33**, 1423-1439 (2019).
- Scott, D. T., Knight, D. M., Blunt-Harris, E. L., Kolesar, S. E., Lovley, D. R., Quinone Moieties Act as Electron
Acceptors in the reduction of Humic Substances by Humics-Reducing Microorganisms, *Environ. Sci. Technol.*,
**32**, 2984-2989 (1998).
- Thorn, K., Folan, D. W., MacCarthy, P., Characterization of the International Humic Substances Society Standard
and Reference fulvic and humic acids by solution state carbon-13 (¹³C) and hydrogen-1 (¹H) nuclear magnetic
resonance spectrometry, US Geological Survey, Water-Resources Investigations Report 89-4196, 1989, pp.1-99.

Zhang, F., Harir, M., Moritz, F., Zhang, J., Witting, M., Wu, Y., Schmitt-Kopplin, P., Fekete, A., Gaspar, A.,
Hertkorn, N. Molecular and structural characterization of dissolved organic matter during and post cyanobacterial
bloom in Taihu by combination of NMR spectroscopy and FTICR mass spectrometry, *Water Research* **57**, 280-
294 (2014).

Reviewer Reports on the First Revision:

Referees' comments:

Referee #1 (Remarks to the Author):

Thanks to the authors for adding clarity to various points raised in the initial review.

Regarding links between Conclusions and data - this is now clearer and the authors have done a good job of recasting this. It is now clear that the data measured (quantitative NMR highlighting the high population of quaternary/tertiary carbons) is being used as evidence that degradation of aromatic compounds is the source of a large component of DOM (rather than a more complex link to particular detailed chemical mechanisms).

However this, for me, reduces the breadth and significance of this paper. While of interest to those working specifically in these fields, it is now not clear to me what the impact of this in a broader context - and particularly why this would not be better placed in a more specialist journal. What can we do/know now that we could not before? A more specialist publication would ensure the audience will appreciate the importance of this evidence, the NMR approach and interpretation used, and be likely to exploit this in future research programs.

Regarding the technical detail of the NMR spectroscopy, the additional control experiments and literature precedent adds clarity and confidence that indeed the relaxation delays are appropriate (I would highlight that 60-120 second relaxation delays are entirely normal in solution-state quantitative ^{13}C NMR experiments, where T_1 of isolated quaternary carbons can be $>10\text{s}$ etc - so it is very system dependent, but it appears DOM presents a less than extreme case here). There is still a small lack of technical detail. For example the reported pulse length is for the 90 degree pulse, but it isn't made clear if/that this is the excitation pulse angle used - which has an enormous effect on how long the relaxation delay needs to be - and so the correlation with literature is still not clear without the value of the excitation pulse angle itself (rather than the calibrated 90 degree pulse). The likelihood is that both values are the same (i.e. a 90 degree excitation angle is being used in the quantitative NMR experiments in the literature and/or in this report) but that is not a given and is not stated.

Referee #2 (Remarks to the Author):

The authors provided thoughtful responses to my earlier comments. The revised article helps clarify the roles of ODA and COOH chemistry in the molecular diversification of DOM. The authors have presented compelling, not conclusive, evidence indicating ODA plays an important role in shaping the chemical complexity of freshwater DOM from the Amazon River system and boreal lakes. This is an important observation that improves understanding of the processes that contribute to the molecular diversity in DOM. These findings and speculations will stimulate research on this topic, which is of critical importance for understanding C cycling and the chemical nature of refractory molecules that persist for centuries to millennia.

The authors present an additional figure (Fig. 4) in the revised manuscript. The figure includes "traditional" and "new" views of the processes driving the chemical complexity of DOM. The figures (4a, 4b) and text are confusing and limited in scope. The "traditional" view (Fig. 4a) should include all recognized processes that contribute to DOM complexity, including biotic and abiotic processes. Biological processes drive chemical diversity, and they include primary production, secondary production, grazing, excretion, exudation, viral lysis, and biodegradation. Other important factors include community structure and promiscuous enzymes. Abiotic processes, including

photochemical and thermal transformations, sorption/desorption, and redox reactions also contribute to the molecular diversity in DOM.

The "new" view (Fig. 4b) presents ODA as an important process that contributes to DOM chemical complexity. The authors propose that ODA and consecutive cascade reactions increase molecular diversity by changing the aromatic structure of polyphenol substrates into complex aliphatic and oxygenated structures. These structures are proposed to be precursors of carboxyl-rich-alicyclic-molecules (CRAM) that are abundant, refractory components of DOM. Figure 4b shows potential ODA and cascade reactions that dearomatize and synthesize a novel molecule that could be a precursor of CRAM. Is this a biotic or abiotic process? The authors don't directly address this question, but I think they should. Do microbes take up "free" polyphenols (<600 Da) in DOM and transform them (ODA-cascade reactions) into oxygenated aliphatic molecules? The synthesis of these molecules would require substantial energy, so it is difficult to understand why microbes would do this. How would these CRAM precursors enter the DOM reservoir? Do abiotic photochemical processes initiate ODA of polyphenols? The authors need to propose how polyphenols in DOM are exposed to ODA and consecutive cascade reactions. The reaction shown in Fig. 4b is taken from Fig. 3e, and it doesn't help explain the processes that transform "free" polyphenols in DOM into oxygenated aliphatic structures. I suggest the authors revise figure 4 to address this conundrum.

Specific comments

Line 71-72: Microbial degradation plays a major role in shaping the molecular composition of boreal lake DOM.

Line 127-129: It is difficult to see in Fig. 2g that N-DOM has the highest proportion of polyphenols. The authors should refer to Fig. 2c to show that N-DOM has the highest proportion of polyphenols.

Line 295: The last sentence is vague and needs clarification. "Unrecognized bioactivity may reside in its molecules."

Line 304-306: The microbial communities responsible for the biodegradation of DOM also vary among ecosystems.

Figure 1a and 1b: Include the abbreviations for the samples collected and analyzed in this study (e.g. S-DOM).

Ron Benner

Referee #1 (Remarks to the Author):

{REDACTED}

Thanks to the authors for adding clarity to various points raised in the initial review.

Regarding links between Conclusions and data - this is now clearer and the authors have done a good job of recasting this. It is now clear that the data measured (quantitative NMR highlighting the high population of quaternary/tertiary carbons) is being used as evidence that degradation of aromatic compounds is the source of a large component of DOM (rather than a more complex link to particular detailed chemical mechanisms).

We are not sure we understand this comment. What we try to highlight is that the oxidative dearomatization reaction (ODA) reaction mechanism shapes the large structural diversity in DOM – not that it is the source of a large component of DOM as said in the comment. Our conclusion is based on the richness of OC_qC₃ quaternary carbon atoms at an abundance that as far as we can see, does not have any other reasonable explanation.

Current science of NOM synthesis offers only chemical reactions which alter a limited number of atomic environments, like. e.g. (de)hydration ($\pm\text{H}_2\text{O}$), (de)hydrogenation ($\pm\text{H}_2$), (de)carbonylation ($\pm\text{CO}$), (de)carboxylation ($\pm\text{CO}_2$), (de)methylation ($\pm\text{CH}_2$), and oxidation/deoxygenation ($\pm\text{O}$). While these reactions primary concern alterations of functional groups in NOM molecules, not any credible reaction has been proposed for rearrangements of carbon skeletons in NOM molecules until now (see new added Extended Data Table 6; cf. below). The overwhelming molecular diversity of NOM molecules so far has precluded proposal of more elaborate organic reaction mechanisms.

ODA activates oxygenated aromatic molecules and makes those subject to extensive rearrangement reactions which simultaneously alter several carbon chemical environments. In fact, the widespread use of ODA in contemporary synthetic organic chemistry resides in its capacity to generate complex molecular shapes from rather simple precursor molecules. Hence, ODA is not at all a simple degradation of oxygenated aromatic molecules like the previously known organic matter degradation reactions. In essence, ODA is the first known chemical reaction of NOM molecules which refers to (extensive) rearrangement of carbon skeletons that includes transformations of several sp²-hybridized carbon into sp³-hybridized carbon. Hence, ODA causes a much more fundamental rearrangement reaction than e.g. the previously described ring opening reaction of benzene derivatives. The first step of this reaction only interchanges sp²-hybridized carbon atoms into another ones (e.g. ring opening

of benzene by dioxygenase enzymes produces isomers of muconic acid, HOOC-CH=CH-CH=CH-COOH which carry only sp²-hybridized carbon atoms).

To further emphasize the distinction between ODA and previously known reactions discussed in the processing of DOM molecules, we have now included Extended Data Table 6 in the Supplementary information. This table specifies the number of carbon atoms affected by the previously discussed main chemical transformations of NOM (entries 1-12), and it contrasts these rather incremental alterations with those effected by ODA, and consecutive cycloadditions (entries 13-16). Here, reactions 1-4 change two sp³-hybridized carbon atoms into two sp²-hybridized carbon atoms, and vice versa. Reactions 5-12 are nominal insertion reactions or its reversal, often referring to C-H bonds (e.g. carboxylation: C-H + CO₂ = C-COO-H) which do not alter atomic environments in the rest of the molecule. In comparison, the combined ODA/cycloaddition reaction transforms five Csp² atomic environments into five Csp³ atomic environments. This combination is the most massive alteration of carbon skeletons in DOM synthesis/degradation ever described. The resulting DOM molecules are heavily fused and bridged alicyclic molecules, with diffuse embedding of oxygen. These features of DOM molecules agree with NMR spectra of freshwater DOM, and cannot be replicated by common biomolecular binding motifs.

Extended Data Table 6 | Chemical reactions in the synthesis and degradation of DOM molecules, and how these affect carbon chemical environments. Reactions 1-12 have been previously described in organic matter research. Oxidative dearomatization (ODA) activates oxygenated aromatic rings in DOM, like e.g. lignins and tannins, making the generated cyclohexadienones subject to consecutive cycloaddition reactions of which the most prominent is of [4+2] type, i.e. Diels-Alder cycloaddition. All reactions 1-16 can be of biotic or/and abiotic origin.

item	reaction	molecular change	change of carbon atomic environments
hydrogenation	+H ₂	2×Csp ² into 2×Csp ³
dehydrogenation	-H ₂	2×Csp ³ into 2×Csp ²
hydration	+H ₂ O	2×Csp ² into 2×Csp ³
dehydration	-H ₂ O	2×Csp ³ into 2×Csp ²
carboxylation	+CO ₂	insertion only, hybridization stays for all other carbon atoms
decarboxylation	-CO ₂	release only, hybridization stays for all other carbon atoms
methylation	+CH ₂	insertion only, hybridization stays for all other carbon atoms
demethylation	-CH ₂	release only, hybridization stays for all other carbon atoms
oxidation (CHO molecules)	+O	insertion only, hybridization stays for all other carbon atoms

reduction (CHO molecules)	-O	release only, hybridization stays for all other carbon atoms
carbonylation	+CO	insertion only, hybridization stays for all other carbon atoms
decarbonylation	-CO	release only, hybridization stays for all other carbon atoms
oxidative dearomatization ODA	molecular composition remains intact (tautomerization)	$1 \times \text{Csp}^2$ into $1 \times \text{Csp}^3$
[4+2] cycloaddition following ODA	molecular mass doubles; elemental ratios remain intact	$4 \times \text{Csp}^2$ into $4 \times \text{Csp}^3$
ODA / general cycloaddition, identical molecules $2 \times A$	molecular mass doubles from A to $2 \times A$; elemental ratios remain intact	$5 \times \text{Csp}^2$ into $5 \times \text{Csp}^3$
ODA / general cycloaddition, distinct molecules A and B	molecular mass increases to A+B; cumulative elemental ratios of educts and products remain intact	$5 \times \text{Csp}^2$ into $5 \times \text{Csp}^3$

However this, for me, reduces the breadth and significance of this paper. While of interest to those working specifically in these fields, it is now not clear to me what the impact of this in a broader context - and particularly why this would not be better placed in a more specialist journal. What can we do/know now that we could not before? A more specialist publication would ensure the audience will appreciate the importance of this evidence, the NMR approach and interpretation used, and be likely to exploit this in future research programs.

At level of facts, this contribution adds previously unknown knowledge about transformations and chemical structures of NOM molecules as follows:

- (1) The first elaborate reaction mechanism of NOM synthesis concerned with carbon skeletal rearrangements.
- (2) First mentioning of aliphatic quaternary carbon atomic environments C_qC_4 and OC_qC_3 in NOM, which convey structural recalcitrance (i.e. longevity) to DOM molecules.
- (3) This reconciles extended lifetime of NOM molecules with their structural features.
- (4) First proof of a major structural distinction between common biomolecules and NOM molecules.

All these novel items will invite further research about molecular mechanisms of NOM transformations in various ecosystems.

However, more relevant from our perspective is the finding that ODA is prevalent in nature. This opens a new door towards understanding organic matter processing in all environments and finally offering knowledge and understanding about not only *why*, but also *how* we find such enormous organic matter diversity in terms of structure and thereby in chemical behavior. This chemical diversity – in turn may have large consequences for carbon storage and cycling in various environments, and for the conditions for and diversity of various life forms, as all life in various ways interact with the organic matter on the planet. Further, it opens doors towards novel use and reshaping of organic matter to compounds of value.

The fact that we happened to discover the prevalence and magnitude of ODA via studies of freshwater DOM does not limit the relevance to this particular context. On the contrary, as our observations stem from studies of highly different source material in very different biomes, it would be very unlikely if ODA is not important in all environments in the biosphere. We believe it is hard to find recent discoveries in chemistry of organic matter with broader relevance.

Regarding the technical detail of the NMR spectroscopy, the additional control experiments and literature precedent adds clarity and confidence that indeed the relaxation delays are appropriate (I would highlight that 60-120 second relaxation delays are entirely normal in solution-state quantitative ^{13}C NMR experiments, where T_1 of isolated quaternary carbons can be $>10\text{s}$ etc - so it is very system dependent, but it appears DOM presents a less than extreme case here). There is still a small lack of technical detail. For example the reported pulse length is for the 90 degree pulse, but it isn't made clear if/that this is the excitation pulse angle used - which has an enormous effect on how long the relaxation delay needs to be - and so the correlation with literature is still not clear without the value of the excitation pulse angle itself (rather than the calibrated 90 degree pulse). The likelihood is that both values are the same (i.e. a 90 degree excitation angle is being used in the quantitative NMR experiments in the literature and/or in this report) but that is not a given and is not stated.

The experimental section of NMR now explicitly states that excitation angle and calibrated 90-degree pulse were identical. This adds clarity and coherence of acquisition across ^{13}C NMR spectra because all acquired ^{13}C NMR spectra used these 90 degree pulses (or 180 degree pulses for spin-echo and DEPT ^{13}C NMR spectra at relevant positions).

Referee #2 (Remarks to the Author):

{REDACTED}

The authors provided thoughtful responses to my earlier comments. The revised article helps clarify the roles of ODA and COOH chemistry in the molecular diversification of DOM. The authors have presented compelling, not conclusive, evidence indicating ODA plays an important role in shaping the chemical complexity of freshwater DOM from the Amazon River system and boreal lakes. This is an important observation that improves understanding of the processes that contribute to the molecular diversity in DOM. These findings and speculations will stimulate research on this topic, which is of critical importance for understanding C cycling and the chemical nature of refractory molecules that persist for centuries to millennia.

The authors fully agree with this statement.

The authors present an additional figure (Fig. 4) in the revised manuscript. The figure includes "traditional" and "new" views of the processes driving the chemical complexity of DOM. The figures (4a, 4b) and text are confusing and limited in scope. The "traditional" view (Fig. 4a) should include all recognized processes that contribute to DOM complexity, including biotic and abiotic processes. Biological processes drive chemical diversity, and they include primary production, secondary production, grazing, excretion, exudation, viral lysis, and biodegradation. Other important factors include community structure and promiscuous enzymes. Abiotic processes, including photochemical and thermal transformations, sorption/desorption, and redox reactions also contribute to the molecular diversity in DOM.

Fig. 4 refers to the structural diversity of DOM molecules during its temporal evolution; it does not distinguish between abiotic and biotic reactions because both will operate in nature to change the chemical structure of DOM molecules during its temporal evolution. This concerns all mentioned reactions (i.e. primary and secondary production, grazing, excretion, exudation, viral lysis, and biodegradation as well as microbial community structure and promiscuous enzymes, and abiotic processes, including photochemical and thermal transformations, sorption/desorption, and redox reactions). We now have moved Fig. 4 to the Supplementary Material, where it is now Extended Data Figure 5. We have changed Extended Data Figure 5b to more clearly address the large increase of structural diversity caused by ODA-related reactions. We also added a more precise figure caption which recognizes the just concerns of the reviewer. To more precisely distinguish between the classic and ODA-based chemical reactions participating in the processing of DOM molecules, we have included new Extended Data Table 6 which mentions the fundamental effects of

traditionally known (items 1-12) and ODA-related chemical reactions in the processing of DOM molecules (items 13-16).

Extended Data Table 6 | Chemical reactions in the synthesis and degradation of DOM molecules, and how these affect carbon chemical environments. Reactions 1-12 have been previously described in organic matter research. Oxidative dearomatization (ODA) activates oxygenated aromatic rings in DOM, like e.g. lignins and tannins, making the generated cyclohexadienones subject to consecutive cycloaddition reactions of which the most prominent is of [4+2] type, i.e. Diels-Alder cycloaddition. All reactions 1-16 can be of biotic or/and abiotic origin.

item	reaction	molecular change	change of carbon atomic environments
hydrogenation	+H ₂	2×Csp ² into 2×Csp ³
dehydrogenation	-H ₂	2×Csp ³ into 2×Csp ²
hydration	+H ₂ O	2×Csp ² into 2×Csp ³
dehydration	-H ₂ O	2×Csp ³ into 2×Csp ²
carboxylation	+CO ₂	insertion only, hybridization stays for all other carbon atoms
decarboxylation	-CO ₂	release only, hybridization stays for all other carbon atoms
methylation	+CH ₂	insertion only, hybridization stays for all other carbon atoms
demethylation	-CH ₂	release only, hybridization stays for all other carbon atoms
oxidation (CHO molecules)	+O	insertion only, hybridization stays for all other carbon atoms
reduction (CHO molecules)	-O	release only, hybridization stays for all other carbon atoms
carbonylation	+CO	insertion only, hybridization stays for all other carbon atoms
decarbonylation	-CO	release only, hybridization stays for all other carbon atoms
oxidative dearomatization ODA	molecular composition remains intact (tautomerization)	1×Csp ² into 1×Csp ³
[4+2] cycloaddition following ODA	molecular mass doubles; elemental ratios remain intact	4×Csp ² into 4×Csp ³

ODA / general cycloaddition, identical molecules $2 \times A$	molecular mass doubles from A to $2 \times A$; elemental ratios remain intact	$5 \times C_{sp^2}$ into $5 \times C_{sp^3}$
ODA / general cycloaddition, distinct molecules A and B	molecular mass increases to $A+B$; cumulative elemental ratios of educts and products remain intact	$5 \times C_{sp^2}$ into $5 \times C_{sp^3}$

The "new" view (Fig. 4b) presents ODA as an important process that contributes to DOM chemical complexity. The authors propose that ODA and consecutive cascade reactions increase molecular diversity by changing the aromatic structure of polyphenol substrates into complex aliphatic and oxygenated structures. These structures are proposed to be precursors of carboxyl-rich-alcyclic-molecules (CRAM) that are abundant, refractory components of DOM. Figure 4b shows potential ODA and cascade reactions that dearomatize and synthesize a novel molecule that could be a precursor of CRAM. Is this a biotic or abiotic process? The authors don't directly address this question, but I think they should.

We had the following wording in our original text after the first revision. Here we stated, corroborated by literature references, that ODA operates by both biotic and abiotic mechanisms, and that its redox-initiated radical, ionic and enzymatic variants show complementary molecular selectivity.

ODA operates through both biotic and abiotic mechanisms^{33,44,47}. Molecular diversification is further
 amplified through dearomatization by complementary selectivity of its photochemical^{48,49}, redox-initiated
 radical^{50,51}, ionic⁵² as well as enzymatic variants: the latter accommodates a remarkable promiscuity of
 substrates⁵³⁻⁵⁵, fostering opportunities for large-scale DOM processing. All these dearomatization reactions
 likely occur in parallel, facilitating rapid complexification within DOM from simple aromatic precursor
 molecules⁴⁵ (Fig. 3, Fig. 4). Fundamentally, ODA transforms flat aromatic rings into elaborately shaped
 oxygenated aliphatic molecules rich in tetrahedral C_{sp^3} -carbon atoms with fused and bridged alicyclic rings
 (Fig. 3e). Aromatic precursor molecules in DOM are of appreciable size (m/z ~450 by mass
 spectrometry)^{20,22}, polysubstituted, polyoxygenated, molecularly diverse, and have inherent low symmetry.
 ODA chemistry of these molecules will inevitably produce highly complex mixtures of oxygen-rich
 alicyclic DOM molecules⁵⁶⁻⁵⁸ (Fig. 3).

Hence, we have nominally addressed this question in our previously submitted manuscript. However, ODA deviates from about all previously discussed molecular transformation of DOM molecules, so that we add some additional

remarks here. There is no space in the main text for such comments because literally hundreds of recent references address both biotic and abiotic ODA reactions as well as their products. We went through the citations of these publications since appearance (date 23.9.2023), many more articles provide evidence for biotic and abiotic ODA reactions which then probably also occur in natural ecosystems. A huge diversity of microorganisms (fungi and bacteria) and plants use ODA to synthesize natural products with elaborate molecular skeletons and a remarkable structural diversity.

article	topic	Citations as of 23.9.2023	article	topic	Citations as of 23.9.2023
33	biotic and abiotic ODA	398	54	flavin enzyme promiscuity	79
44	biotic and abiotic ODA	15	55	flavin enzyme promiscuity	10
biotic and abiotic ODA	946	45	transformation of simple aromatic molecules	12
photochemical ODA	7	20	mass spectrometry of DOM	41
photochemical ODA	46	21	mass spectrometry of DOM	37
redox and radical ODA	1084	56	oxygen-rich alicyclic molecules	22
redox and radical ODA	323	57	oxygen-rich alicyclic molecules	89
ionic ODA	2855	58	oxygen-rich alicyclic molecules	41
flavin enzyme promiscuity	18

To be more specific than possible in the main text, we want to highlight the synthetic potential of flavin-dependent monooxygenases which often mediate highly site- and stereoselective ODA of phenolic substrates. Here, many wild-type biocatalysts demonstrate a remarkable promiscuity of substrates. Recent investigations have uncovered a diverse collection of enzymes from bacterial, fungal, and plant sources capable of generating and directing regio-, stereo-, and chemoselective reactions via o-quinone methide intermediates in aqueous solution (cf. Fig. 3b; Purdy TN et al., Harnessing ortho-Quinone Methides in Natural Product Synthesis and Biocatalysis, *J. Nat. Prod.* **85** (2022) 688-701). Streptomyces bacteria synthesize a range of meroterpenoids by means of ODA reactions (George JH, Biomimetic Dearomatization Strategies in the Total Synthesis of Meroterpenoid Natural Products, *Acc. Chem. Res.*, **54** (2021) 1843-1855).

Overall, ODA-related processing and attendant structural evolution of DOM molecules, will attenuate the classical focus on the fate of initially abundant but labile substrates sugars, lipids and peptides, which may contribute less to the chemical structures of evolved, "old" DOM molecules than currently anticipated.

Do microbes take up "free" polyphenols (<600 Da) in DOM and transform them (ODA-cascade reactions) into oxygenated aliphatic molecules? The synthesis of these molecules would require substantial energy, so it is difficult to understand why microbes would do this. How would these CRAM precursors enter the DOM reservoir? Do abiotic photochemical processes initiate ODA of polyphenols? The authors need to propose how polyphenols in DOM are exposed to ODA and consecutive cascade reactions. The reaction shown in Fig. 4b is taken from Fig. 3e, and it doesn't help explain the processes that transform "free" polyphenols in DOM into oxygenated aliphatic structures. I suggest the authors revise figure 4 to address this conundrum.

ODA-related synthesis of natural products by microorganisms, plants and abiotic reactions is well documented, and sophisticated biosynthetic machinery evolved over billions of years for their production (Huck, CJ et al., Dearomative logic in natural product total synthesis, *Nat. Prod. Rep.*, **39** (2022) 2231-2291; doi: 10.1039/d2np00042c). For answering this question, I want to provide a personal account, summarizing a few dozens of recent publications. Given the high current research activity, the given selection concerning opportunities for biological ODA reactions cannot be exhaustive. I will not specifically mention abiotic (e.g. photochemical) ODA reactions here more than once, because these are well documented, and will probably apply under natural conditions as well. Photocatalytic ODA provides environmentally benign entry to highly substituted o-quinols, reactive intermediates which can be elaborated to a number of natural product families (Baker Dockrey et al., Photocatalytic Oxidative Dearomatization of Orcinaldehyde Derivatives, *Org. Lett.*, **22** (2020) 3712-3716; DOI: 10.1021/acs.orglett.0c01207).

Microbes may take up small phenolic molecules itself (note that ODA with consecutive cycloaddition increases the mass of participating molecules A and B to new mass A+B; see new Extended Data Table 6), or/and substituted oxygenated aromatic rings (substituted aromatic flat rings are sterically accessible for e.g. Flavin-dependent monooxygenases (FMOs) with their large active centers, even when these units are part of larger molecules; cf. below). An ample supply of extracellular oxygenase enzymes is recorded for many microorganisms (e.g. Xu Z et al., Understanding of bacterial lignin extracellular degradation mechanisms by *Pseudomonas putida* KT2440 via secretomic analysis, *Biotechnol. Biofuels Bioprod.* **15** (2022) 117; about 5% of extracellular enzymes described here are monooxygenases).

First, resonant electron donation by oxygenated substituents destabilizes aromatic rings (i.e. decreases their aromaticity, often considerably). Here, phenolate anions are even less aromatic than phenols as demonstrated by (computed) charge densities in o-, m- p- positions for phenols and phenolates, i.e. ortho: 1.008/1.027, meta: 1.012/1.042, para: 1.000/1.130 (Hight RJ and Hight PF, The characterization of complex phenols by nuclear magnetic resonance spectra, *J. Org. Chem.* **30** (1965) 902-906). ¹H NMR chemical shifts of alkyl- and alkoxy substituted phenols and phenolate anions also reflect the

higher electron density at ortho and para positions in case of phenolate anions (D₂O/DMSO): ortho: 0.16-0.28/0.42-0.59 ppm, meta: 0.05-0.11/0.19-0.38 ppm, para: 0.32-0.39/0.71-0.79 ppm upfield ¹H NMR chemical shift. It is interesting that in the case of Flavin-dependent Monooxygenase (FMO) TropB, the active form of phenol subject to ODA reaction is a phenolate anion (Benitez, AR, et al., Structural Basis for Selectivity in Flavin-Dependent Monooxygenase-Catalyzed Oxidative Dearomatization, *ACS Catalysis*, **9** (2019) 3633-3640). Phenolate anions also possess a pronounced tendency for single-electron transfer (SET) reactions, as the resulting phenoxyl radicals are well stabilized by electron delocalization, opening further avenues for abiotic ODA reactions. Already this supply of energy facilitates tautomerization of oxygenated benzene derivatives into cyclohexadienones (Fig. 3a). Facile oxidation to quinones, ketalization of quinones, and formation of quinone methides furnish reactive cyclohexadienones under many environmentally plausible conditions (Fig. 3ab). The chemical energy stored in cyclohexadienones makes them susceptible to fast cycloaddition, and often cascade reactions. This particularly applies to the highly reactive ortho-cyclohexadienones (cf. Fig. 3ab, top row), and ortho-quinone methides. Recent investigations have uncovered a diverse collection of enzymes from bacterial, fungal, and plant sources capable of generating and directing regio-, stereo-, and chemoselective reactions via ortho-quinone methide intermediates. Remarkably, these enzymatic reactions take place under aqueous conditions traditionally avoided due to the nucleophilicity of water promoting rapid quenching of o-QMs (Purdy, TN, et al., Harnessing ortho-Quinone Methides in Natural Product Biosynthesis and Biocatalysis, *J. Nat. Prod.*, **85** (2022) 688-701).

During processing of freshwater DOM molecules in the environment, ODA is possibly not one of the most early reactions of DOM molecules, but it is anticipated that reactive substrates like sugars, lipids and amino acids become exhausted rather quickly under suitable conditions, making alternative reactions like e.g. ODA becoming more competitive. For example, the biosynthesis of complex meroterpenoids of mixed terpene and aromatic polyketide origin often involves cascade reactions that are prompted by dearomatization of an electron-rich aromatic ring (George JH, Biomimetic Dearomatization Strategies in the Total Synthesis of Meroterpenoid Natural Products, *Acc. Chem. Res.*, **54** (2021) 1843-1855).

In my view, (promiscuous) Flavin-dependent monooxygenases feature peculiar features to facilitate widespread environmentally mediated ODA reactions, including large active centers capable of accommodating a considerable range of substrates, and a diversity of oxidation reactions. A recent investigation of the substrate scope and scalability of a panel of flavin-dependent monooxygenases that mediate ODA reactions in various secondary metabolite pathways revealed that in contrast to traditional chemical transformations in which site-selectivity is governed by the substrate, selectivity in the flavin-dependent monooxygenase reaction is dictated by the orientation of the substrate in the active site relative to the oxidant, C4a-hydroperoxyflavin. Benitez et al. envisioned that a suite of flavin-dependent monooxygenases (FMO) could provide complementary selectivity and redefine the state-of-the-art method for mediating this transformation (Benitez, AR, et al., Structural Basis for Selectivity in Flavin-Dependent Monooxygenase-Catalyzed Oxidative Dearomatization, *ACS Catalysis*, **9** (2019) 3633-3640). The crystal structure of TropB revealed a spacious active site capable to accommodate a wide a range of substrate, and site- and

stereoselectivity across an array of substrates through changing relative orientation of reaction partners.

More than 135 FMOs have been described in 2022 (Huijbers, MME, et al., Flavin dependent monooxygenases, *Arch. Biochem. Biophys.*, **544** (2014) 2-17; this article was cited 339 times in 28.9.2023). FMOs are catalytically intricate enzymes. Catalysis depends on a prosthetic group (the FAD), a co-substrate (molecular oxygen), an electron donor (NADPH), the substrate and the protein matrix (Bailleul, G et al., Evolution of enzyme functionality in the flavin-containing monooxygenases, *Nat. Commun.*, **14** (2023) 1042). For FMOs, functional diversity is caused by a network of residues outside the active site, that modulates the formation of the oxygenating flavin intermediate. Group A FMOs are widely involved in the microbial degradation of (poly)aromatic compounds, and opportunities for overlap between more classical aromatic oxidation and ODA-type reactions exist. Novel reactivity of these enzymes is continually unearthed (Teufel et al., Flavin-mediated dual oxidation controls an enzymatic Favorskii-type rearrangement, *Nature*, **503** (2013) 552-556), in part expanding their classification beyond the current group A-G flavin-dependent monooxygenases.

Specific comments

Line 71-72: Microbial degradation plays a major role in shaping the molecular composition of boreal lake DOM.

We have added this important mechanism.

Line 127-129: It is difficult to see in Fig. 2g that N-DOM has the highest proportion of polyphenols. The authors should refer to Fig. 2c to show that N-DOM has the highest proportion of polyphenols.

This note is correct; we have changed accordingly, and referred to Fig. 2c; Fig. 2g shows relatively higher abundance of polyphenols within the specified limited section of chemical shift (δ_c : 90-165 ppm)

Line 295: The last sentence is vague and needs clarification. "Unrecognized bioactivity may reside in its molecules."

We have changed this (originally vague) sentence to = It is conceivable that some of these polyfunctional, elaborately shaped, compact molecules carry relevant but yet unrecognized bioactivity.

Line 304-306: The microbial communities responsible for the biodegradation of DOM also vary among ecosystems.

This is an important point, and we have changed the text accordingly.

Figure 1a and 1b: Include the abbreviations for the samples collected and analyzed in this study (e.g. S-DOM).

Thank you for pointing this out; we have changed Fig. 1a and 1b accordingly.

Ron Benner

Reviewer Reports on the Second Revision:

Referees' comments:

Referee #1 (Remarks to the Author):

I thank the authors for their detailed response, and can only sympathise at the challenges of the data/email losses earlier in the year. I hope this hasn't delayed their wider work too badly.

Regarding the manuscript, the authors have described in their rebuttal a clear explanation of their thinking, which aligns to my understanding of the updated manuscript - that ODA is a source of quaternary structural complexity in these naturally derived molecules in DOM. ODA is well established to do this in biosynthetic systems - including in references used in this manuscript e.g. 42 - hence why I stated in my earlier review that the significance of this work appears narrower in breadth, highlighting this to those working in DOM and environmental research may provide new insight in those realms, but is not a new chemical principle for mechanisms of creating structural complexity in nature and I still hold the view that a more specialist journal would be appropriate for this elegant work.

Referee #2 (Remarks to the Author):

The authors carefully and fully addressed my previous questions and comments. They present compelling and exciting data indicating ODA is a key process in shaping the molecular diversity of DOM in aquatic environments. This observation provides novel and important insights about the C cycle. It appears that ODA is an important mechanism to produce structurally altered DOM molecules that resist degradation and persist in the environment for centuries to millennia. ODA appears to be a key process in the formation of carboxyl-rich alicyclic molecules (CRAM). Several studies have demonstrated that CRAM is abundant in freshwater and the global ocean. It has been shown that CRAM in the deep ocean is old and very resistant to microbial and photochemical degradation. The article would be of broader interest and greater significance if the authors expanded on these independent observations in the Conclusions section of the manuscript. The sequestration of C in CRAM reduces the release of carbon dioxide to the atmosphere and thereby impacts global warming and climate change.

Abstract (some suggestions to reduce word count)

Line 26 - delete "mechanisms within" and replace with "in"

Line 33 - delete "massively"

Line 34 - delete "drastic"

Line 35 - delete "inflationary increase of mainly aliphatic structures with abundant oxygenation" and replace with "an increase in oxygenated aliphatic structures"

Line 36-38 - change "the most credible" to "a key", delete "massive", delete ", and probably prevalent in nature" and replace with "in nature".

Similar editing of the manuscript would reduce the overall word count. The number of references could also be reduced by selecting only the most important articles for citation.

Fig. 1. The maps of DOM sampling sights are not needed in the manuscript, and they could be moved to the Extended Figures section. The first Extended Figure is a more detailed map of sampling sites in the Amazon, so Figure 1 can be integrated in Extended Figure 1.

Fig. 2. Figure 2G shows spectra from 60-165 ppm, not 90-165 ppm.

Table 1 legend – revise as follows: “Percentages of eleven ^{13}C NMR-derived key carbon chemical environments ($\text{CH}_0\text{123}$) in DOM samples. Csp^3 -based quaternary carbon (Cq) units are shaded (i.e. carbon not carrying any hydrogen according to NMR notation). Ipso-CarC refers to 1,3,5-trioxo-polyphenols²⁵.”

Overall, the number of references can be reduced by choosing the most important articles for citation.

The Extended Figures and Tables are very informative and helpful for understanding ODA chemistry and NMR analyses.

Ron Benner